# PMLF: A Physics-Guided Multiscale Loss Framework for Structurally Heterogeneous Time Series

**Xinghong Chen,** **Weilin Wu** *, **Kungping Yang,** **Guannan Chen** *

Key Laboratory of OptoElectronic Science and Technology for Medicine of Ministry of Education
Fujian Normal university
qbx20220087@yjs.fjnu.edu.cn; weilin_wu1990@163.com;
kungpingyang@fjnu.edu.cn; edado@fjnu.edu.cn

## Abstract

Forecasting real-world time series requires modeling both short-term fluctuations and long-term evolutions, as these signals typically exhibit multiscale temporal structures. A core challenge lies in reconciling such dynamics: high-frequency seasonality demands local precision, while low-frequency trends require global robustness. However, most existing methods adopt a unified loss function across all temporal components, overlooking their structural differences. This misalignment often causes overfitting to seasonal noise or underfitting of long-term trends, leading to suboptimal forecasting performance. To address this issue, we propose a Physics-guided Multiscale Loss Framework (PMLF) that decomposes time series into seasonal and trend components and assigns component-specific objectives grounded in the distinct energy responses of oscillatory and drift dynamics. Specifically, we assign a quadratic loss to seasonal components, reflecting the quadratic potential energy profile of molecular vibration, while a logarithmic loss is used for trend components to capture the sublinear energy profile of molecular drift under sustained external forces. Furthermore, we introduce a softmax-based strategy that adaptively balances the unequal energetic responses of these two physical processes. Experiments on different public benchmarks show that PMLF improves the performance of diverse baselines, demonstrating the effectiveness of physics-guided loss design in modeling structural heterogeneity in time series forecasting.

## 1 Introduction

Time series forecasting is widely used across a range of scientific and industrial domains[34], including energy consumption [35], climate change [40, 43], financial exchange [24], and traffic flow [20]. Real-world time series inherently exhibit complex multiscale behavior, combining high-frequency seasonal variation with low-frequency trend dynamics. This multiscale characteristic introduces conflicting demands on forecasting models, as short-term and long-term patterns evolve at different temporal resolutions, making accurate forecasting particularly challenging[30]. Recent methods address multiscale forecasting primarily through architectural-level innovations, such as hierarchical attention[22], multi-resolution convolutions [29], seasonal-trend decomposition[39, 38], and frequency-aware representations [45, 41, 5, 28]. While these approaches improve the model's ability to represent multiscale signals, they apply a uniform loss functions during optimization, with limited attention paid to how different temporal structures are supervised [3, 15, 10].

Figure 1 illustrates the divergent error behaviors of seasonal and trend components. Seasonal components are highly sensitive to minor phase shifts or amplitude distortions, which rapidly accumulate over time and significantly degrade predictive accuracy. In contrast, trend components

---

*Corresponding Author: Guannan Chen, Weilin Wu

are more tolerant to local errors but may suffer from long-term drift when trained under overly strict constraints. These differences indicate that seasonal and trend components respond to supervision in fundamentally different ways and thus require structurally distinct loss formulations [36, 33]. However, most existing approaches adopt a single loss, such as MSE or MAE, applied indiscriminately across the entire sequence [13, 17, 2]. This uniform treatment implicitly assumes that all temporal components contribute equally to forecasting error. As a result, the model fails to distinguish between structurally different error patterns, which limits its ability to adapt supervision to each component and ultimately impairs forecasting accuracy and long-term stability.

To resolve the mismatch in optimization objectives between seasonal and trend components in multiscale time series, we draw inspiration from structural response mechanisms in molecular systems [37]. Specifically, we design loss functions based on the energy profiles characteristic of each component. Molecular vibrations follow harmonic dynamics around equilibrium, where small displacements lead to rapid increases in potential energy [7]. In contrast, molecular structures under sustained external influence exhibit nonlinear relaxation, with weakening restoring forces and sublinear energy accumulation[25]. These two physical responses closely parallel the behaviors of seasonal and trend components in time series.

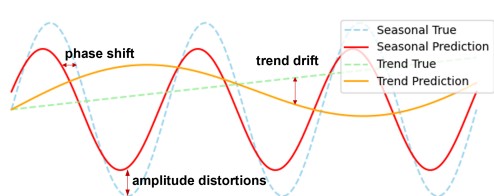

Figure 1: Illustration of distinct error patterns arising from different structural components in time series. Seasonal components are particularly sensitive to phase shifts and amplitude distortions, whereas trend components are more susceptible to long-term drift.

Motivated by this analogy, we propose a Physics-guided Multiscale Loss Framework (PMLF) that supervises each component using a loss function grounded in its corresponding physical dynamics. As illustrated in Figure 2, once the model produces the output sequence, both the prediction and the ground truth are decomposed during loss computation into seasonal and trend components, each supervised by a structurally consistent objective. The seasonal branch is optimized using a quadratic loss, consistent with harmonic energy, while the trend branch adopts a logarithmic loss that reflects sublinear stress accumulation under long-term drift. To ensure balanced training across heterogeneous error profiles, we further incorporate a softmax-based weighting mechanism that dynamically adjusts the relative contributions of seasonal and trend losses. This loss design yields the following contributions:

- We reveal that seasonal and trend components exhibit fundamentally heterogeneous structural characteristics, analogous to harmonic oscillations and irreversible structural drift in physical systems.

- Based on this insight, we formulate a physics-guided loss framework that assigns quadratic penalties to seasonal errors and logarithmic potentials to trend deviations, reflecting their distinct structural behaviors. We further integrate a softmax-based weighting mechanism to dynamically balance their learning contributions during training.

- Experiments on different public benchmarks show that PMLF improves the performance of diverse baselines, demonstrating the effectiveness of physics-guided loss design in modeling structural heterogeneity in time series forecasting.

## 2 Related Work

### 2.1 Multiscale Modeling in Time Series Forecasting

Due to the inherent decomposability of time series, many recent forecasting models have taken multiscale modeling strategies, including explicit temporal decomposition, frequency-domain decomposition, and hierarchical temporal abstraction. Explicit decomposition methods, such as Autoformer [38], extract trend signals via moving-average filtering and treat seasonal variations as residual components. TimeMixer [29] extends this idea by operating over multiple temporal resolutions to better capture fine-grained temporal structures. Frequency-domain approaches model signals in the spectral domain to identify dominant periodicities and multiscale frequency structures, as

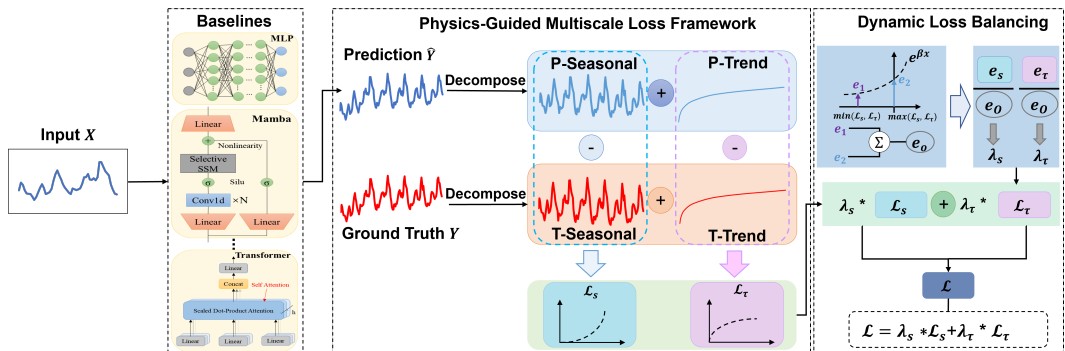

Figure 2: Overview of the proposed Physical-Guided Multiscale Loss Framework. Given an input time series, the model predicts future values, and both predictions and ground truths are decomposed into seasonal and trend components. Each component is supervised by a physically grounded loss aligned with its structural behavior. Finally, a softmax-based dynamic weighting mechanism then balances their contributions to form the final optimization objective.

demonstrated by FEDformer [45], FreTS[42], and WPMixer[21]. Hierarchical models [17, 18, 8] instead organize inputs into layered or patch-wise representations to capture dependencies across temporal scales. PatchTST[22], for example, achieves strong performance by applying self-attention to fixed-length patches, enabling joint modeling of short- and long-term dynamics. Although these models incorporate multiscale structures at the architectural level, they typically adopt uniform loss functions across all components. This mismatch between representation and supervision may limit the model's ability to learn the distinct dynamics of trend and seasonal signals.

## 2.2 Loss Function Design in Time Series Forecasting

Beyond point-wise error minimization, recent work has explored shape-aware loss functions designed to improve sequence-level accuracy. These approaches can be broadly categorized into alignment-based and structure-aware objectives. Alignment-based losses, including DTW [1], Soft-DTW [3], DILATE [15], and TIDLE-Q [16], enable elastic matching between predictions and targets, improving robustness to temporal misalignment and phase variation. Structure-aware losses, in contrast, aim to preserve internal signal characteristics such as frequency, locality, and temporal coherence. For example, FreDF [27] compares signals in the frequency domain to preserve spectral energy distributions. Patch-wise structural loss [11] learns local correlation, variance, and mean patterns by decomposing sequences into overlapping patches, enabling finer-grained structural supervision.

# 3 Methodology

## 3.1 Overview and Motivation

While recent forecasting models incorporate structural decomposition to isolate trend and seasonal components, their training objectives typically remain uniform, applying loss functions such as MSE over the full output sequence. However, this uniform supervision fails to reflect the divergent dynamic sensitivities of multiscale components. Localized seasonal oscillations require fine-grained alignment, whereas long-term trends benefit from stable, drift-aware objectives.

To address this mismatch between representation and supervision, we propose a physics-guided multiscale loss framework. As illustrated in Figure 2, both predictions and ground truths are decomposed into seasonal and trend components using a shared operator. Each component is then supervised by a loss function aligned with its temporal dynamics. This structure-aware supervision allows the model to optimize multiscale representations more effectively, leading to improved long-range forecasting performance.

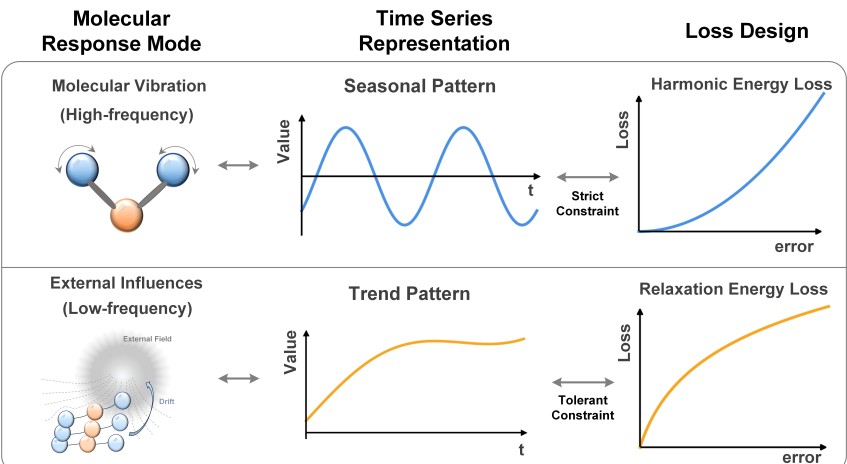

Figure 3: Conceptual illustration of our physics-guided loss design. Seasonal components are analogous to high-frequency molecular vibrations and are penalized using harmonic energy loss, which imposes strict local alignment. Trend components resemble structural drift under external fields and are optimized via a relaxation energy loss that enforces tolerant, sublinear correction.

## 3.2 Problem Formulation

Given a multivariate time series $X = \{x_1, x_2, ..., x_T\} \in \mathbb{R}^{T \times C}$, the model $f_\theta$ aims to predict the next $H$ time steps conditioned on a historical window of length $L$:

$$\hat{Y} = \{\hat{y}_{T+1}, ..., \hat{y}_{T+H}\} = f_\theta(x_{T-L+1}, ..., x_T) \qquad (1)$$

To capture multiscale temporal structure, the prediction and ground truth sequences are decomposed into two components via seasonal-trend decomposition:

$$Y = Y_\tau + Y_s, \quad \hat{Y} = \hat{Y}_\tau + \hat{Y}_s \qquad (2)$$

where $Y_\tau$ and $Y_s$ denote the trend and seasonal component, respectively. This decomposition is performed using a shared low-pass filter $\mathcal{D}$ to extract the trend, while the seasonal component is defined as the residual. We omit the stochastic noise term in this work, as it lacks structured patterns and thus cannot be effectively supervised.

## 3.3 Physics-Inspired Structural Supervision

In molecular systems, physical responses to perturbations generally fall into two categories: intrinsic vibrations and externally induced structural drift [37, 23, 9]. As shown in Figure 3, the former corresponds to reversible, high-frequency oscillations of atomic bonds around equilibrium, exhibiting strong sensitivity to small displacements. The latter arises from persistent external forces such as thermal gradients or mechanical stress, leading to irreversible, low-frequency drift with gradually diminishing resistance. These two physical response modes naturally correspond to the structural decomposition of time series. Seasonal components exhibit periodic fluctuations and are highly sensitive to phase and amplitude deviations, similar to molecular vibrations. In contrast, trend components evolve gradually under long-term influence and resemble structural drift observed in dissipative systems. To supervise these components in a structure-consistent manner, we replace uniform loss penalties with energy-based objectives whose growth profiles align with the dynamic behavior of each component.

**Energy-Based Loss for Seasonal Oscillation.**   In molecular systems, small displacements from equilibrium are governed by harmonic motion, where the potential energy increases quadratically with deviation:

$$U_{vib}(x) = \frac{1}{2}k(x - x_0)^2 \qquad (3)$$

where $x$ is the system state, $x_0$ the equilibrium position, and $k$ the stiffness constant. This response structure reflects the behavior of seasonal time series components, which exhibit high-frequency, reversible oscillations and are sensitive to phase or amplitude shifts.

Given predicted and target seasonal values $\hat{s}(t)$ and $s(t)$, the potential energy difference is:

$$\Delta U_{vib}(t) = U_{vib}(\hat{s}(t)) - U_{vib}(s(t)) = \frac{k}{2}\left[\hat{s}(t)^2 - s(t)^2 - 2x_0(\hat{s}(t) - s(t))\right] \tag{4}$$

By expanding the square and reorganizing terms, the energy difference can be rewritten as:

$$\Delta U_{vib}(t) = \frac{k}{2}\left[(\hat{s}(t) - s(t))^2 - 2(s(t) - x_0)(\hat{s}(t) - s(t))\right] \tag{5}$$

Due to the symmetric oscillation of $s(t)$ around its equilibrium, the coupling term $(s(t) - x_0)(\hat{s}(t) - s(t))$ has an expected value close to zero across a complete seasonal cycle:

$$\mathbb{E}_t\left[(s(t) - x_0)(\hat{s}(t) - s(t))\right] \approx 0 \tag{6}$$

Consequently, the expected energy difference simplifies to a dominant quadratic penalty, which defines the seasonal loss as:

$$\mathcal{L}_s = \frac{1}{CH}\sum_{c,h}\left(\hat{s}_{T+h}^{(c)} - s_{T+h}^{(c)}\right)^2 \tag{7}$$

**Relaxation Energy Loss for Trend Components.** In contrast to the symmetric and reversible oscillations of seasonal dynamics, trend components reflect irreversible structural drift induced by persistent external influences. Similar dynamics occur in dissipative physical systems under sustained forces, such as thermal gradients, where deformation accumulates progressively and the system's internal opposition to change gradually saturates. To capture this non-restorative response, we introduce a diminishing structural force model:

$$r(e) = \frac{ke}{1 + \alpha e}, \quad \text{where } e = \hat{\tau} - \tau \tag{8}$$

which captures the saturation of structural resistance under persistent displacement.

The integral of this response yields the effective structural energy potential:

$$U_{rel}(e) = \frac{k}{\alpha^2}\left(1 + \alpha e - \log(1 + \alpha e)\right) \tag{9}$$

which reflects the sublinear accumulation of internal structural energy associated with irreversible drift. Linear and constant terms are omitted as they yield non-informative or structurally insensitive gradients. We retain only the logarithmic penalty, which preserves structural sensitivity and yields the final trend loss:

$$\mathcal{L}_\tau = \frac{1}{CH}\sum_{c,h}\log\left(1 + \left|\hat{\tau}_{T+h}^{(c)} - \tau_{T+h}^{(c)}\right|\right) \tag{10}$$

## 3.4 Dynamic Loss Balancing

Seasonal and trend components originate from distinct physical mechanisms: harmonic oscillation and structural relaxation, respectively. When supervised jointly, their associated energy responses evolve at different rates. As training proceeds, the trend-related loss often grows several orders of magnitude larger than the seasonal loss due to its slow but persistent accumulation. This numerical imbalance distorts the force equilibrium between the two components, causing the trend branch to dominate gradient updates and suppressing high-frequency corrections. To address this, we adopt a softmax-based dynamic weighting scheme that adjusts the influence of each component based on its current energy level. Instead of static weights, we compute adaptive coefficients from the exponentially scaled differences between the detached losses. This strategy maintains structural balance during optimization and avoids second-order effects.

The total loss is defined as a weighted sum:

$$\mathcal{L} = \lambda_s\mathcal{L}_s + \lambda_\tau\mathcal{L}_\tau \tag{11}$$

Table 1: Multivariate long-term forecasting results across diverse real-world datasets. The table reports MSE and MAE across four prediction horizon $H \in \{96, 192, 336, 720\}$, with a fixed input sequence length of 96. The better results for each setting are highlighted in bold.

| Method | | Amplifier[5](2025) | | | | TimeXer[31](2024) | | | | S-Mamba(2024)[32] | | | | iTransformer[19](2024) | | | | TimeMixer[29](2024) | | | |
|---|---|---|---|---|---|---|---|---|---|---|---|---|---|---|---|---|---|---|---|---|---|
| Loss Functions | | MSE | | PMLF | | MSE | | PMLF | | MSE | | PMLF | | MSE | | PMLF | | MSE | | PMLF | |
| Metrics | | MSE | MAE | MSE | MAE | MSE | MAE | MSE | MAE | MSE | MAE | MSE | MAE | MSE | MAE | MSE | MAE | MSE | MAE | MSE | MAE |
| ETTh1 | 96 | 0.385 | 0.398 | **0.375** | **0.394** | 0.390 | 0.402 | **0.380** | **0.402** | 0.387 | 0.406 | **0.386** | **0.404** | 0.390 | 0.407 | **0.387** | **0.403** | 0.385 | 0.397 | **0.365** | **0.392** |
| | 192 | 0.448 | 0.44 | **0.423** | **0.425** | 0.443 | 0.433 | **0.424** | **0.430** | 0.445 | 0.441 | **0.437** | **0.433** | 0.448 | 0.441 | **0.438** | **0.434** | 0.435 | 0.426 | **0.416** | **0.422** |
| | 336 | **0.484** | **0.45** | 0.499 | 0.461 | 0.475 | 0.452 | **0.462** | **0.443** | 0.495 | 0.470 | **0.477** | **0.452** | 0.485 | 0.459 | **0.479** | **0.453** | 0.49 | 0.455 | **0.464** | **0.445** |
| | 720 | 0.539 | 0.499 | **0.484** | **0.467** | 0.488 | **0.479** | **0.480** | 0.481 | 0.505 | 0.496 | **0.479** | **0.477** | 0.494 | 0.484 | **0.486** | **0.478** | 0.512 | 0.493 | **0.471** | **0.464** |
| | Avg | 0.464 | 0.447 | **0.443** | **0.436** | 0.449 | 0.442 | **0.437** | **0.439** | 0.458 | 0.453 | **0.445** | **0.442** | 0.454 | 0.448 | **0.448** | **0.442** | 0.456 | 0.443 | **0.429** | **0.431** |
| ETTh2 | 96 | 0.303 | 0.355 | **0.283** | **0.332** | 0.285 | 0.336 | **0.283** | **0.332** | 0.294 | 0.346 | **0.290** | **0.337** | 0.300 | 0.351 | **0.296** | **0.340** | **0.297** | 0.346 | 0.297 | **0.337** |
| | 192 | 0.369 | 0.397 | **0.353** | **0.379** | 0.368 | 0.392 | **0.361** | **0.381** | 0.378 | 0.397 | **0.375** | **0.391** | 0.381 | 0.399 | **0.372** | **0.390** | 0.375 | 0.395 | **0.365** | **0.381** |
| | 336 | 0.412 | 0.428 | **0.385** | **0.405** | 0.419 | 0.428 | **0.392** | **0.409** | 0.419 | 0.432 | **0.399** | **0.411** | 0.408 | 0.425 | **0.414** | **0.421** | 0.426 | 0.428 | **0.383** | **0.403** |
| | 720 | 0.446 | 0.455 | **0.403** | **0.423** | 0.423 | 0.441 | **0.415** | **0.431** | 0.433 | 0.451 | **0.402** | **0.424** | 0.422 | 0.442 | **0.412** | **0.430** | 0.432 | 0.445 | **0.402** | **0.427** |
| | Avg | 0.383 | 0.409 | **0.356** | **0.385** | 0.374 | 0.399 | **0.363** | **0.388** | 0.381 | 0.407 | **0.367** | **0.391** | 0.378 | 0.404 | **0.374** | **0.395** | 0.383 | 0.404 | **0.362** | **0.387** |
| ETTm1 | 96 | 0.316 | 0.355 | **0.311** | **0.342** | 0.324 | 0.361 | **0.321** | **0.349** | 0.370 | 0.391 | **0.329** | **0.356** | 0.344 | 0.377 | **0.326** | **0.353** | 0.334 | 0.367 | **0.316** | **0.349** |
| | 192 | **0.364** | 0.385 | 0.371 | **0.372** | 0.376 | 0.389 | **0.369** | **0.373** | 0.384 | 0.398 | **0.379** | **0.351** | 0.382 | 0.393 | **0.378** | **0.378** | 0.368 | 0.384 | **0.367** | **0.378** |
| | 336 | **0.396** | 0.406 | 0.399 | **0.395** | 0.409 | 0.411 | **0.405** | **0.399** | 0.45 | 0.435 | **0.420** | **0.411** | 0.419 | 0.417 | **0.412** | **0.402** | 0.398 | 0.405 | **0.395** | **0.401** |
| | 720 | **0.460** | 0.446 | 0.481 | **0.436** | 0.467 | 0.443 | **0.463** | **0.432** | 0.523 | 0.478 | **0.473** | **0.441** | 0.487 | 0.455 | **0.477** | **0.439** | 0.452 | 0.44 | **0.453** | **0.436** |
| | Avg | **0.384** | 0.398 | 0.390 | **0.386** | 0.394 | 0.401 | **0.390** | **0.388** | 0.432 | 0.426 | **0.400** | **0.390** | 0.408 | 0.411 | **0.398** | **0.393** | 0.388 | 0.399 | **0.383** | **0.391** |
| ETTm2 | 96 | 0.180 | 0.262 | **0.172** | **0.25** | 0.172 | 0.258 | **0.168** | **0.246** | 0.195 | 0.278 | **0.180** | **0.260** | 0.188 | 0.274 | **0.175** | **0.252** | 0.176 | 0.259 | **0.170** | **0.249** |
| | 192 | 0.241 | 0.301 | **0.237** | **0.293** | 0.237 | 0.300 | **0.233** | **0.290** | 0.256 | 0.314 | **0.243** | **0.299** | 0.252 | 0.312 | **0.243** | **0.296** | 0.238 | 0.299 | **0.236** | **0.293** |
| | 336 | 0.303 | 0.344 | **0.299** | **0.332** | 0.298 | 0.339 | **0.294** | **0.329** | 0.327 | 0.356 | **0.303** | **0.337** | 0.314 | 0.351 | **0.305** | **0.336** | 0.301 | 0.341 | 0.302 | **0.335** |
| | 720 | 0.399 | 0.399 | **0.391** | **0.387** | 0.394 | 0.396 | **0.394** | **0.388** | 0.416 | 0.407 | **0.401** | **0.394** | 0.413 | 0.406 | **0.402** | **0.393** | 0.393 | 0.395 | **0.390** | **0.388** |
| | Avg | 0.281 | 0.327 | **0.275** | **0.316** | 0.275 | 0.323 | **0.272** | **0.313** | 0.299 | 0.339 | **0.282** | **0.323** | 0.292 | 0.336 | **0.281** | **0.319** | 0.277 | 0.324 | **0.275** | **0.316** |
| Weather | 96 | 0.164 | 0.209 | **0.156** | **0.193** | 0.157 | 0.205 | **0.156** | **0.195** | 0.165 | 0.210 | **0.161** | **0.199** | 0.174 | 0.214 | **0.168** | **0.202** | 0.163 | 0.209 | **0.162** | **0.198** |
| | 192 | 0.213 | 0.253 | **0.209** | **0.243** | 0.204 | 0.247 | **0.202** | **0.237** | 0.214 | 0.252 | **0.208** | **0.242** | 0.221 | 0.254 | **0.219** | **0.247** | **0.208** | 0.25 | **0.208** | **0.242** |
| | 336 | 0.268 | 0.292 | **0.262** | **0.283** | 0.261 | 0.290 | **0.260** | **0.280** | 0.274 | 0.297 | **0.263** | **0.284** | 0.278 | 0.296 | **0.272** | **0.289** | 0.251 | 0.287 | 0.263 | **0.284** |
| | 720 | 0.344 | 0.342 | **0.339** | **0.334** | 0.340 | 0.341 | **0.336** | **0.332** | 0.350 | 0.345 | **0.342** | **0.335** | 0.358 | 0.347 | **0.351** | **0.341** | 0.339 | 0.341 | 0.339 | **0.336** |
| | Avg | 0.247 | 0.274 | **0.242** | **0.263** | 0.241 | 0.271 | **0.239** | **0.261** | 0.251 | 0.276 | **0.244** | **0.265** | 0.258 | 0.278 | **0.253** | **0.270** | **0.240** | 0.272 | 0.243 | **0.265** |
| ECL | 96 | 0.152 | 0.248 | **0.148** | **0.242** | 0.140 | 0.242 | **0.138** | **0.236** | 0.139 | 0.235 | **0.137** | **0.232** | 0.148 | 0.240 | **0.146** | **0.236** | **0.158** | 0.251 | 0.158 | **0.245** |
| | 192 | 0.162 | 0.256 | **0.161** | **0.243** | 0.157 | 0.256 | **0.155** | **0.251** | 0.159 | 0.255 | 0.161 | **0.254** | **0.162** | 0.253 | 0.163 | **0.252** | 0.171 | 0.260 | **0.170** | **0.259** |
| | 336 | 0.174 | 0.269 | **0.170** | **0.263** | 0.176 | 0.275 | **0.171** | **0.269** | 0.176 | 0.272 | **0.175** | **0.268** | 0.178 | 0.269 | **0.173** | **0.264** | 0.188 | 0.280 | **0.187** | **0.275** |
| | 720 | 0.203 | 0.294 | **0.199** | **0.288** | 0.211 | 0.306 | **0.201** | **0.293** | 0.204 | 0.298 | **0.203** | **0.293** | 0.225 | 0.317 | **0.204** | **0.292** | 0.228 | 0.314 | **0.227** | **0.308** |
| | Avg | 0.172 | 0.267 | **0.170** | **0.259** | 0.171 | 0.270 | **0.166** | **0.262** | 0.170 | 0.265 | **0.169** | **0.262** | 0.178 | 0.270 | **0.172** | **0.261** | 0.186 | 0.277 | **0.185** | **0.272** |
| Traffic | 96 | 0.455 | **0.298** | **0.451** | **0.298** | **0.428** | 0.271 | 0.429 | **0.268** | **0.382** | 0.261 | 0.385 | **0.258** | 0.395 | 0.268 | **0.393** | **0.255** | **0.462** | 0.285 | 0.470 | 0.297 |
| | 192 | 0.470 | 0.316 | **0.467** | **0.296** | 0.448 | 0.282 | 0.453 | **0.279** | 0.396 | 0.267 | **0.394** | **0.265** | 0.417 | 0.276 | **0.412** | **0.263** | **0.473** | 0.296 | 0.491 | 0.304 |
| | 336 | 0.479 | 0.316 | **0.472** | **0.314** | **0.473** | 0.289 | 0.483 | **0.286** | **0.417** | 0.276 | 0.428 | **0.274** | 0.433 | 0.283 | **0.428** | **0.270** | **0.498** | 0.296 | 0.516 | 0.314 |
| | 720 | 0.523 | **0.328** | **0.511** | 0.331 | 0.516 | 0.307 | 0.521 | **0.303** | 0.460 | 0.300 | **0.450** | **0.295** | 0.467 | 0.302 | **0.461** | **0.288** | **0.506** | 0.313 | 0.531 | 0.328 |
| | Avg | 0.482 | 0.315 | **0.475** | **0.310** | **0.466** | 0.287 | 0.472 | **0.284** | 0.414 | 0.276 | **0.412** | **0.273** | 0.428 | 0.282 | **0.424** | **0.269** | **0.485** | 0.298 | 0.499 | 0.31 |
| Solar | 96 | 0.189 | 0.222 | **0.188** | **0.214** | 0.189 | 0.276 | **0.187** | **0.241** | 0.205 | 0.244 | **0.202** | **0.212** | 0.203 | 0.237 | **0.199** | **0.218** | **0.189** | 0.259 | 0.2 | **0.232** |
| | 192 | 0.225 | 0.256 | **0.221** | **0.231** | 0.210 | 0.295 | **0.201** | **0.257** | 0.237 | 0.270 | **0.235** | **0.239** | **0.233** | 0.261 | 0.238 | **0.244** | **0.222** | 0.283 | 0.224 | **0.254** |
| | 336 | **0.245** | 0.265 | 0.245 | **0.248** | 0.215 | 0.299 | **0.213** | **0.267** | 0.258 | 0.288 | **0.248** | **0.255** | 0.248 | 0.273 | 0.258 | **0.258** | **0.231** | 0.292 | 0.243 | **0.266** |
| | 720 | **0.253** | 0.278 | 0.253 | **0.251** | 0.230 | 0.313 | **0.220** | **0.272** | 0.260 | 0.288 | **0.251** | **0.256** | **0.249** | 0.275 | 0.256 | **0.255** | **0.223** | 0.285 | 0.243 | **0.278** |
| | Avg | 0.228 | 0.255 | **0.227** | **0.236** | 0.211 | 0.296 | **0.205** | **0.259** | 0.240 | 0.273 | **0.234** | **0.241** | **0.233** | 0.262 | 0.238 | **0.244** | **0.216** | 0.280 | 0.228 | **0.256** |

where the adaptive weights $\lambda_s$ and $\lambda_\tau$ are computed as:

$$\lambda_s = \frac{\exp\big(\beta(\mathcal{L}_s^\dagger - m)\big)}{\exp\big(\beta(\mathcal{L}_s^\dagger - m)\big) + \exp\big(\beta(\mathcal{L}_\tau^\dagger - m)\big)}, \quad \lambda_\tau = \frac{\exp\big(\beta(\mathcal{L}_\tau^\dagger - m)\big)}{\exp\big(\beta(\mathcal{L}_s^\dagger - m)\big) + \exp\big(\beta(\mathcal{L}_\tau^\dagger - m)\big)} \quad (12)$$

Here, $\mathcal{L}^\dagger = \text{SG}(\mathcal{L})$ denotes the gradient-detached loss to avoid backpropagating through the weighting mechanism. The scalar $m = \max(\mathcal{L}_s^\dagger, \mathcal{L}_\tau^\dagger)$ ensures numerical stability, and the parameter $\beta = 1$ unless otherwise specified.

# 4 Experiments

## 4.1 Setup

**Datasets** To comprehensively evaluate the effectiveness of PMLF on real datasets, we conducted experiments on eight multivariate time series datasets, including ETT (4 sub-datasets) [44], Weather, Electricity, Traffic, and Solar-Energy [12]. Detailed dataset descriptions are provided in Appendix.

**Baselines** To evaluate the ability of PMLF on different baselines, we extensively selected Transformer, Mamba, MLP, CNN, and their hybrid methods, including Amplifier [5], TimeXer [31], S-Mamba [32], iTransformer [19], TimeMixer [29], PatchTST [22], and TimesNet [39].

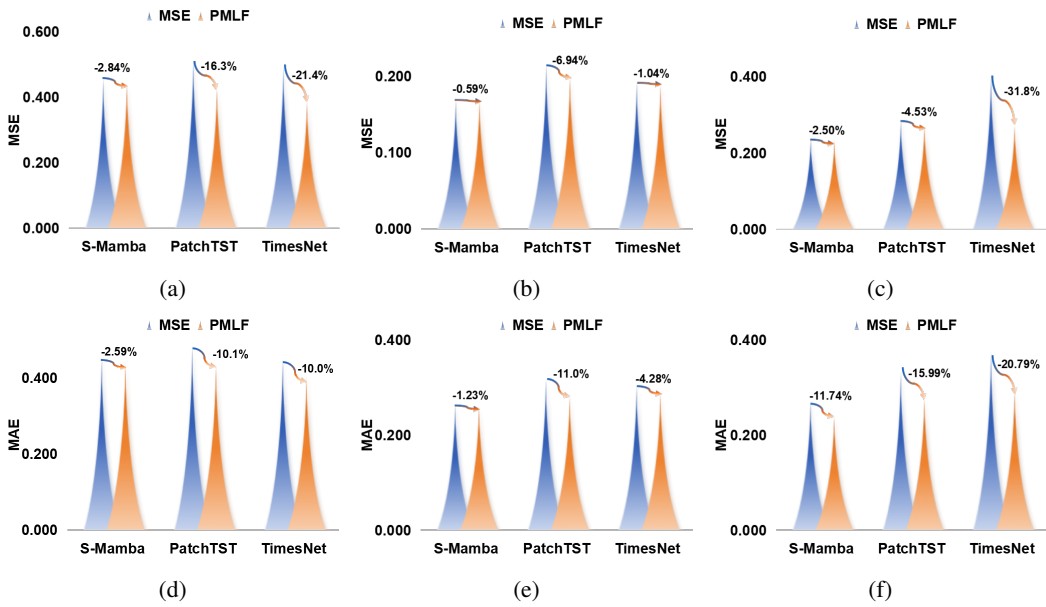

Figure 4: Forecasting performance of three backbone models (S-Mamba, PatchTST, and TimesNet) on the ETTh1, ECL, and Solar datasets. Subplots (a–c) show MSE and (d–f) show MAE, averaged over four forecasting horizons $H \in \{96, 192, 336, 720\}$. Each pair of bars compares standard MSE loss with the proposed PMLF.

**Implementation Detail** All models were trained from scratch using PMLF as the loss function, ensuring a consistent evaluation of its effectiveness. During training, seasonal and trend components are extracted via a fixed moving average filter with kernel size 25, and we also support an optional learnable version implemented as a plug-in module. Our loss framework is model-agnostic and can be seamlessly integrated into a wide range of forecasting backbones, including MLPs [5], Transformers [26], and Mamba [6, 4], thereby supporting universal applicability to multiscale prediction tasks. To ensure fair comparison, we ensure the consistency of the parameters of the comparison loss, only by adjusting the learning rate to adapt to the gradient dynamics caused by our loss design. All experiments are conducted using PyTorch with four NVIDIA RTX 4090 GPUs.

## 4.2 Main Results

We evaluate the proposed PMLF loss across a set of state-of-the-art forecasting models on eight benchmark datasets covering diverse application domains. The standard MSE loss is used as the baseline for comparison. As shown in Table 1, the proposed model-agnostic loss framework yields consistent improvements over the standard MSE baseline across five recent state-of-the-art models. This demonstrates that PMLF effectively guides the optimization of both short-term seasonal patterns and long-term trends, which frequently coexist in real-world time series. Especially, the performance improvement of the model on the MAE index is often higher than that on the MSE index, mainly due to the long-term control of trend. For example, among a total of 64 comparisons in iTransformer, the PMLF improvement item accounted for 93.75% (60/64). In addition, the improvement in the MAE metric is often more substantial than that in MSE, suggesting that the trend-aware loss contributes significantly to long-term stability. To further assess the generalizability of our framework, Figure 4 reports the average forecasting performance of three representative architectures: S-Mamba, PatchTST, and TimesNet, across multiple prediction horizons. On typical datasets including ETTh1, ECL, and Solar-Energy, PMLF achieves significant improvements in both MSE and MAE. Additional results covering all models and datasets are provided in the appendix.

## 4.3 Comparison with Other Loss Functions

We compare PMLF with traditional loss function and specially designed objectives proposed for time series forecasting. These approaches typically enhance standard objectives by introducing additional

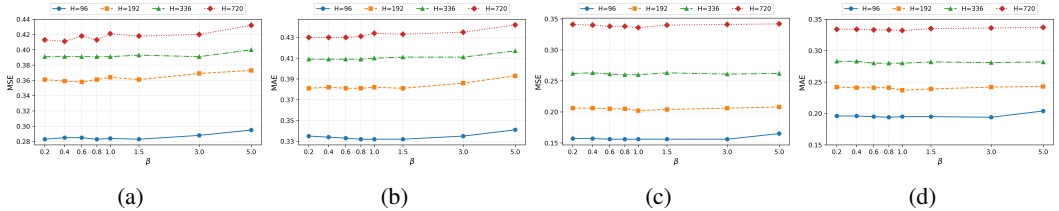

(a)         (b)         (c)         (d)

Figure 5: Robustness analysis of the dynamic weighting parameter $\beta$ on the ETTh2 (a-b) and Weather (c-d) datasets based on TimeXer.

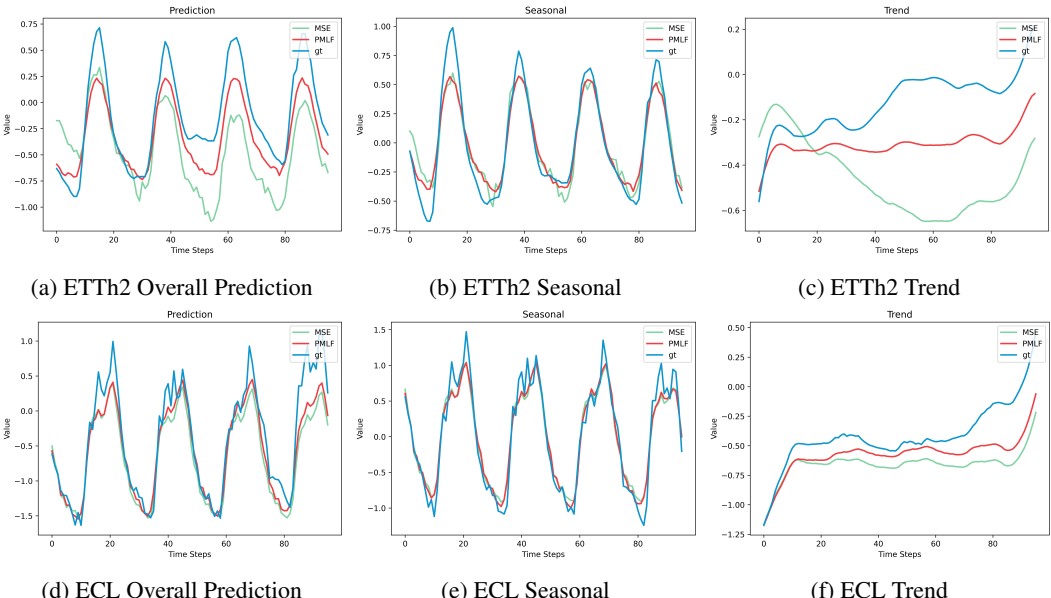

(a) ETTh2 Overall Prediction     (b) ETTh2 Seasonal     (c) ETTh2 Trend

(d) ECL Overall Prediction     (e) ECL Seasonal     (f) ECL Trend

Figure 6: **Overall and Component-wise Prediction Visualization on ETTh2 and ECL.** Comparison between MSE and PMLF losses on ETTh2 ((a)–(c)) and ECL ((d)–(f)). Each row presents the overall prediction, trend component, and seasonal component. PMLF exhibits closer alignment with the ground truth in both local seasonal and global trends.

terms to emphasize specific learning characteristics. For instance, TILDE-Q [16] introduces penalties on phase and amplitude shifts to improve temporal alignment, while FreDF strengthens label-wise consistency through frequency-domain comparisons. In contrast, PMLF [27] supervises decomposed seasonal and trend components individually at the loss level, aiming to guide structural learning rather than refining a unified temporal loss. As shown in Table 2, PMLF achieves better overall performance on three datasets in both MSE and MAE metrics. These results demonstrate that structural supervision introduced solely through the loss function can effectively improve forecasting accuracy.

## 4.4 Visualization

To assess the structural impact of our loss function, we visualize predictions on ETTh2 and ECL using Amplifier as the backbone model. As shown in Figure 6, compared to the standard MSE loss, our proposed PMLF loss produces forecasts that are more closely aligned with the ground truth. To further analyze the effect of structural supervision, we decompose both predicted and true sequences into seasonal and trend components. In both dimensions, the PMLF-based predictions exhibit improved alignment with their respective ground truth counterparts, demonstrating enhanced fidelity in modeling high-frequency oscillations as well as long-term drift.

Table 2: Comparison of PMLF with standard and structure-aware loss functions on ETTh2, ETTm2, and Weather datasets using Amplifier as the backbone model.

| Dataset | | ETTh2 | | | | ETTm2 | | | | Weather | | | |
|---|---|---|---|---|---|---|---|---|---|---|---|---|---|
| horizon | | 96 | 192 | 336 | 720 | 96 | 192 | 336 | 720 | 96 | 192 | 336 | 720 |
| MSE | MSE | 0.303 | 0.369 | 0.412 | 0.446 | 0.180 | 0.241 | 0.303 | 0.399 | 0.164 | 0.213 | 0.268 | 0.344 |
| | MAE | 0.355 | 0.397 | 0.428 | 0.455 | 0.262 | 0.301 | 0.344 | 0.399 | 0.209 | 0.268 | 0.292 | 0.342 |
| Soft-DTW | MSE | 0.292 | 0.371 | 0.413 | 0.436 | 0.177 | 0.242 | 0.302 | 0.407 | 0.164 | 0.214 | 0.268 | 0.344 |
| | MAE | 0.346 | 0.396 | 0.429 | 0.450 | 0.259 | 0.303 | 0.344 | 0.406 | 0.209 | 0.252 | 0.291 | 0.341 |
| Huber | MSE | 0.298 | 0.360 | 0.399 | 0.432 | 0.174 | 0.238 | 0.299 | 0.397 | 0.159 | 0.209 | 0.265 | 0.342 |
| | MAE | 0.345 | 0.388 | 0.420 | 0.444 | 0.255 | 0.299 | 0.339 | 0.396 | 0.201 | 0.246 | 0.287 | 0.338 |
| TILDE-Q [16] | MSE | 0.286 | 0.364 | 0.394 | 0.420 | 0.177 | 0.236 | 0.296 | 0.390 | 0.169 | 0.214 | 0.267 | 0.345 |
| | MAE | 0.334 | 0.384 | 0.419 | 0.438 | 0.254 | 0.293 | 0.333 | 0.390 | 0.213 | 0.250 | 0.291 | 0.341 |
| FreDF [27] | MSE | 0.285 | 0.355 | 0.392 | 0.422 | 0.173 | **0.235** | **0.296** | **0.388** | 0.167 | 0.212 | 0.269 | 0.345 |
| | MAE | 0.334 | 0.381 | **0.412** | 0.436 | 0.253 | 0.294 | 0.334 | 0.388 | 0.210 | 0.254 | 0.297 | 0.348 |
| PMLF | MSE | **0.283** | **0.353** | **0.39** | **0.403** | **0.172** | 0.237 | 0.299 | 0.391 | **0.156** | **0.209** | **0.262** | **0.339** |
| | MAE | **0.332** | **0.379** | 0.417 | **0.423** | **0.251** | **0.293** | **0.332** | **0.387** | **0.193** | **0.243** | **0.283** | **0.334** |

Table 3: Ablation study of the components of PMLF loss on the ETTh2, ETTm2 and Weather datasets using Amplifier as a backbone.

| Method | | PMLF | | w/o $\mathcal{L}_s$ | | w/o $\mathcal{L}_\tau$ | | w/o Balancing | | w/ LMA | |
|---|---|---|---|---|---|---|---|---|---|---|---|
| Metirc | | MSE | MAE | MSE | MAE | MSE | MAE | MSE | MAE | MSE | MAE |
| ETTh2 | 96 | **0.283** | **0.332** | 0.330 | 0.361 | 0.375 | 0.391 | 0.292 | 0.335 | 0.289 | **0.332** |
| | 192 | **0.353** | **0.379** | 0.487 | 0.468 | 0.428 | 0.423 | 0.357 | 0.382 | 0.357 | **0.379** |
| | 336 | **0.385** | **0.405** | 0.670 | 0.588 | 0.421 | 0.429 | 0.392 | 0.409 | 0.391 | 0.406 |
| | 720 | **0.403** | **0.423** | 0.592 | 0.548 | 0.434 | 0.446 | 0.406 | 0.425 | 0.406 | 0.425 |
| ETTm2 | 96 | 0.172 | 0.251 | 0.206 | 0.277 | 0.208 | 0.282 | 0.175 | 0.256 | **0.171** | **0.250** |
| | 192 | **0.237** | **0.293** | 0.301 | 0.345 | 0.249 | 0.303 | 0.239 | 0.297 | 0.239 | 0.295 |
| | 336 | 0.299 | 0.332 | 0.348 | 0.369 | 0.308 | 0.340 | 0.301 | 0.338 | **0.296** | **0.331** |
| | 720 | **0.391** | **0.387** | 0.433 | 0.414 | 0.405 | 0.396 | 0.400 | 0.392 | 0.393 | 0.390 |
| Weather | 96 | 0.156 | **0.193** | 0.21 | 0.262 | 0.180 | 0.223 | 0.158 | 0.197 | **0.155** | 0.194 |
| | 192 | **0.209** | **0.243** | 0.259 | 0.292 | 0.217 | 0.254 | 0.212 | 0.249 | **0.209** | **0.243** |
| | 336 | **0.262** | **0.283** | 0.306 | 0.323 | 0.271 | 0.292 | 0.265 | 0.286 | 0.264 | **0.283** |
| | 720 | **0.339** | **0.334** | 0.367 | 0.360 | 0.346 | 0.341 | 0.343 | 0.335 | 0.342 | **0.334** |

## 4.5 Ablation Studies

As shown in Table 3, we conduct ablation studies on the ETTh2, ETTm2, and Weather datasets using Amplifier as the backbone. The results show that removing either the seasonal or trend component from the loss formulation leads to a significant drop in forecasting accuracy, confirming the necessity of jointly modeling both temporal structures. Additionally, eliminating the dynamic weighting mechanism by fixing the loss coefficients also degrades performance, highlighting the importance of adaptive balancing between the two structural modes. Furthermore, we implement a learnable moving average (LMA) module with trainable kernels. Its performance remains comparable to the fixed-kernel version, suggesting that the proposed loss is compatible with both static and adaptive decomposition strategies. Finally, as illustrated in Figure 7, we assess the robustness of the hyperparameter $\beta$ on the ETTh1 and Weather datasets using the TimeXer model. The results indicate that the performance remains stable across a wide range of $\beta$ values, from 0.2 to 5, demonstrating that our loss design is insensitive to this choice.

# 5 Conclusion

We introduce a Physics-Guided Multiscale Loss Framework (PMLF) for time series forecasting, which decouples seasonal and trend supervision by grounding each in distinct physical dynamics. Inspired by molecular systems, the framework models short-term oscillations through harmonic potentials and long-term drift via relaxation energy. Furthermore, a softmax-based dynamic weighting mechanism balances the contributions of each component during training, enabling adaptive optimization across structural error profiles. Experimental results across multiple datasets and model backbones show that PMLF consistently improves forecasting accuracy.

***Limitation and Future Works.*** However, PMLF assumes that multiscale structures are present and separable. Although this assumption holds for many real-world signals, the benefits of structural decoupling may diminish in settings dominated by abrupt transitions or highly regular periodic signals, such as regime shifts or physiological rhythms like heartbeat and respiration. These signals lack the multi scale heterogeneity that PMLF is designed to exploit. A promising direction for future work is to explore adaptive decomposition mechanisms or hybrid supervision strategies that enhance the framework's applicability to structure-invariant or single-scale time series.

## Acknowledgements

This study was supported by the Natural Science Foundation of Fujian Province under Grant 2024J01063 and Grant 2024Y4017.

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

# A Datasets and Implementation

## A.1 Datasets

We conducted experiments on 8 real-world datasets to evaluate the effectiveness of the our proposed PMLF loss functions across various domains. The detailed dateset information are depicted in Figure 4

- **ETT**(Electricity Transformer Temperature): The ETT dataset contains 7 variables of electricity transformer temperature from July 2016 to july 2018. There are 4 sub datasets: ETTh1, ETTh2, ETTm2, ETTm2, where ETTh recorded hourly and ETTm recorded every 15 minutes.

- **Weather**: Weather contains 21 meteorological variables collected every 10 minutes from the Weather Station of the Max Planck Biogeochemistry Institute in 2020.

- **ECL** : ECL records the hourly electricity consumption data of 321 clients from 2012 to 2014.

- **Traffic**: Traffic collects hourly road occupancy rates measured by 862 sensors of San Francisco Bay area freeways from January 2015 to December 2016.

- **Solar-Energy**: Solar records the solar power production of 137 PV plants in 2006, which are sampled every 10 minutes.

We follow the same data processing and train-validation-test set split protocol used in TimesNet, where the train, validation, and test datasets are strictly divided according to chronological order to make sure there are no data leakage issues. As for the forecasting settings, we fix the length of the lookback series as 96 and the prediction length varies in {96, 192, 336, 720}.

Table 4: Detailed Dataset Descriptions. Dim denotes the variable number of each dataset. Prediction Length denotes the future time steps to be predicted and four prediction setting are included in each dataset. Dataset Size denotes the total number of time steps in (Train, Validation, Test) split respectively. Frequency denotes the sampling interval of time steps.

| Dataset | Dim | Prediction Length | Dataset Size | Frequency | Domain |
|---|---|---|---|---|---|
| ETTh1 | 7 | {96, 192, 336, 720} | (8545, 2881, 2881) | 1 hour | Electricity |
| ETTh2 | 7 | {96, 192, 336, 720} | (8545, 2881, 2881) | 1 hour | Electricity |
| ETTm1 | 7 | {96, 192, 336, 720} | (34465, 11521, 11521) | 15 min | Electricity |
| ETTm2 | 7 | {96, 192, 336, 720} | (34465, 11521, 11521) | 15 min | Electricity |
| Weather | 21 | {96, 192, 336, 720} | (36792, 5271, 10540) | 10 min | Weather |
| ECL | 321 | {96, 192, 336, 720} | (18317, 2633, 5261) | 1 hour | Electricity |
| Traffic | 862 | {96, 192, 336, 720} | (12185, 1757, 3509) | 1 hour | Transportation |
| Solar-Energy | 137 | {96, 192, 336, 720} | (36601, 5161, 10417) | 10 min | Energy |

## A.2 Implementation Details

All the experiments are implemented in PyTorch and conducted on four NVIDIA 4090 24GB GPU. For fair comparison, we set the input size for all models to be uniform, with the batchsize for ETT and Weather datasets set to 64 and the ECL, Traffic, and Solar datasets set to 16. Except for changing the learning rate to fully learn new structures, do not change other parameters related to the model.

Before calculating the loss, the time series is decomposed into seasonal and trend components using the moving average method. The kernel size refers to the setting in Autoformer, which is 25. But in order to avoid fixed kernel sizes affecting time series with different sampling frequencies, we designed a hybrid expert decomposition mechanism that uses a set of average pooling layers to extract trends and combines them with learnable weights. The kernel sizes {7, 13, 15, 25, 49} represent the corresponding periods at different frequencies.

# B    Baselines

To evaluate the general applicability of PMLF, we compare it with a varied collection of leading time-series forecasters that span the principal architectural families: state-space (S-Mamba), Transformer (iTransformer, TimeXer, PatchTST), multilayer perceptron (Amplifier, TimeMixer), and convolutional neural network (TimesNet). The core ideas of these baselines are outlined below.

- Amplifier: An energy-amplification block heightens weak spectral bands, the spectrum is re-normalised, and parallel seasonal and trend heads with a lightweight channel-interaction module enhance performance on low-signal datasets.

- TimeXer: Targets mixed endogenous and exogenous forecasting. Patch-wise self-attention models the target series, variate-wise cross-attention injects exogenous cues, and global endogenous tokens integrate the two streams, improving robustness to abrupt external shocks.

- S-Mamba: Represents each time step with a single per-channel token and applies a bidirectional Mamba state-space layer along the channel axis to capture inter-variable dependencies. This near-linear-time design surpasses Transformer baselines while greatly reducing computational cost.

- iTransformer: Reassigns Transformer roles by applying attention across variables to learn cross-channel links, while the feed-forward block operates along time to model nonlinear temporal dynamics. This rearrangement lowers memory usage for long horizons and yields interpretable variable-level attention.

- TimeMixer: A pure-MLP predictor. Depth-wise convolutions extract multi-scale bands, linear layers mix these components, and a parallel head simultaneously outputs all future steps. The absence of attention provides GPU-friendly speed without sacrificing accuracy.

- PatchTST: Divides long sequences into fixed-length temporal patches that serve as Transformer tokens and shares encoder parameters among channels. This strategy reduces attention complexity from $\mathcal{O}(L^2)$ to $\mathcal{O}((L/P)^2)$ while retaining local semantics.

- TimesNet: Converts a one-dimensional series into a two-dimensional time-period grid and employs heterogeneous CNN kernels to capture intra-period seasonality as well as inter-period trends. This representation enables direct transfer from vision backbones and delivers strong results in forecasting, anomaly detection, and classification.

These heterogeneous baselines ensure that any improvements attributed to PMLF are not limited to a single modelling philosophy but instead reflect a broad enhancement of time-series learning.

# C    More Experimental Results

## C.1    Robustness Assessment

To examine the robustness of our framework, the Amplifier baseline was trained five times with independent random seeds. Table 5 reports the mean performance together with the corresponding standard deviations. The consistently small variances confirm that the Amplifier yields repeatable results, underscoring the robustness of the proposed approach.

## C.2    Parameter Sensitivity

To examine how the dynamic-weighting coefficient $\beta$ influences forecasting accuracy, we conducted a grid search over $\beta \in \{0.2, 0.4, 0.6, 0.8, 1, 1.5, 3, 5\}$ for two representative networks: TimeXer and TimeMixer. Figure 7 reports the resulting MSE and MAE on the ETTh2 and Weather datasets. For TimeXer (top row) and TimeMixer (bottom row), the error curves remain nearly flat across the entire range, and the optimal $\beta$ values cluster around 1.0 on both datasets. The maximum deviation from the best MSE and MAE is below 2.5%, indicating that the proposed dynamic weighting scheme is insensitive to the precise choice of $\beta$. These results confirm the robustness of our framework with respect to this hyper-parameter.

Table 5: Roubustness of PMLF performance. The results are obtained from five random seeds using the Amplifier as backbone.

| Dataset | ETTh1 | | ETTh2 | | Weather | |
|---|---|---|---|---|---|---|
| Horizon | MSE | MAE | MSE | MAE | MSE | MAE |
| 96 | 0.375±0.002 | 0.394±0.001 | 0.283±0.000 | 0.332±0.000 | 0.156±0.001 | 0.193±0.001 |
| 192 | 0.423±0.003 | 0.425±0.002 | 0.353±0.002 | 0.379±0.001 | 0.209±0.002 | 0.243±0.004 |
| 336 | 0.490±0.003 | 0.456±0.002 | 0.385±0.000 | 0.405±0.001 | 0.262±0.002 | 0.283±0.004 |
| 720 | 0.484±0.003 | 0.467±0.002 | 0.403±0.002 | 0.423±0.001 | 0.339±0.002 | 0.334±0.004 |
| Dataset | ETTm1 | | ETTm2 | | ECL | |
| Horizon | MSE | MAE | MSE | MAE | MSE | MAE |
| 96 | 0.311±0.002 | 0.342±0.002 | 0.172±0.002 | 0.250±0.001 | 0.148±0.000 | 0.242±0.000 |
| 192 | 0.371±0.004 | 0.372±0.003 | 0.237±0.001 | 0.293±0.002 | 0.161±0.001 | 0.243±0.000 |
| 336 | 0.399±0.003 | 0.395±0.003 | 0.299±0.002 | 0.332±0.002 | 0.170±0.000 | 0.263±0.001 |
| 720 | 0.481±0.005 | 0.436±0.004 | 0.391±0.003 | 0.387±0.004 | 0.199±0.001 | 0.288±0.001 |

(a) TimeXer:ETTh1  (b) TimeXer:ETTh1  (c) TimeXer:Weather  (d) TimeXer:Weather

(e) TimeMixer:ETTh1  (f) TimeMixer:ETTh1  (g) TimeMixer:Weather  (h) TimeMixer:Weather

Figure 7: Sensitivity of the dynamic-weighting coefficient $\beta$. The first row shows TimeXer performance on ETTh2 (a, b) and Weather (c, d) as $\beta$ varies, and the second row gives the corresponding results for TimeMixer.

## C.3 Additional Evaluation on Classical Forecasting Networks

Additionally, we assesses the architecture-independent effectiveness of PMLF on three widely used forecasting models: the Transformer-based PatchTST, the selective state-space model S Mamba, and the convolutional network TimesNet. For each architecture, the original MSE objective was replaced with PMLF and the models were evaluated on the ETT, Weather, ECL, and Solar datasets under four prediction horizons $\{96, 192, 336, 720\}$. As shown in Figure 8, across all datasets and horizons, the substitution of MSE with PMLF consistently reduced the error metrics (MAE, MSE), confirming that PMLF improves forecasting accuracy and robustness even in classical network settings that are not covered by the main set of state-of-the-art architectures.

## D Visualization

As shown in Figure 9 and 10, we provide a visual comparison between PMLF and MSE on six benchmark datasets: ETTh2, ETTm2, Weather, ECL, Traffic, and Solar. Each row corresponds to one dataset, where the first column presents the overall forecast, and the second and third columns show the decomposed seasonal and trend components, respectively. Across all datasets, the forecasts

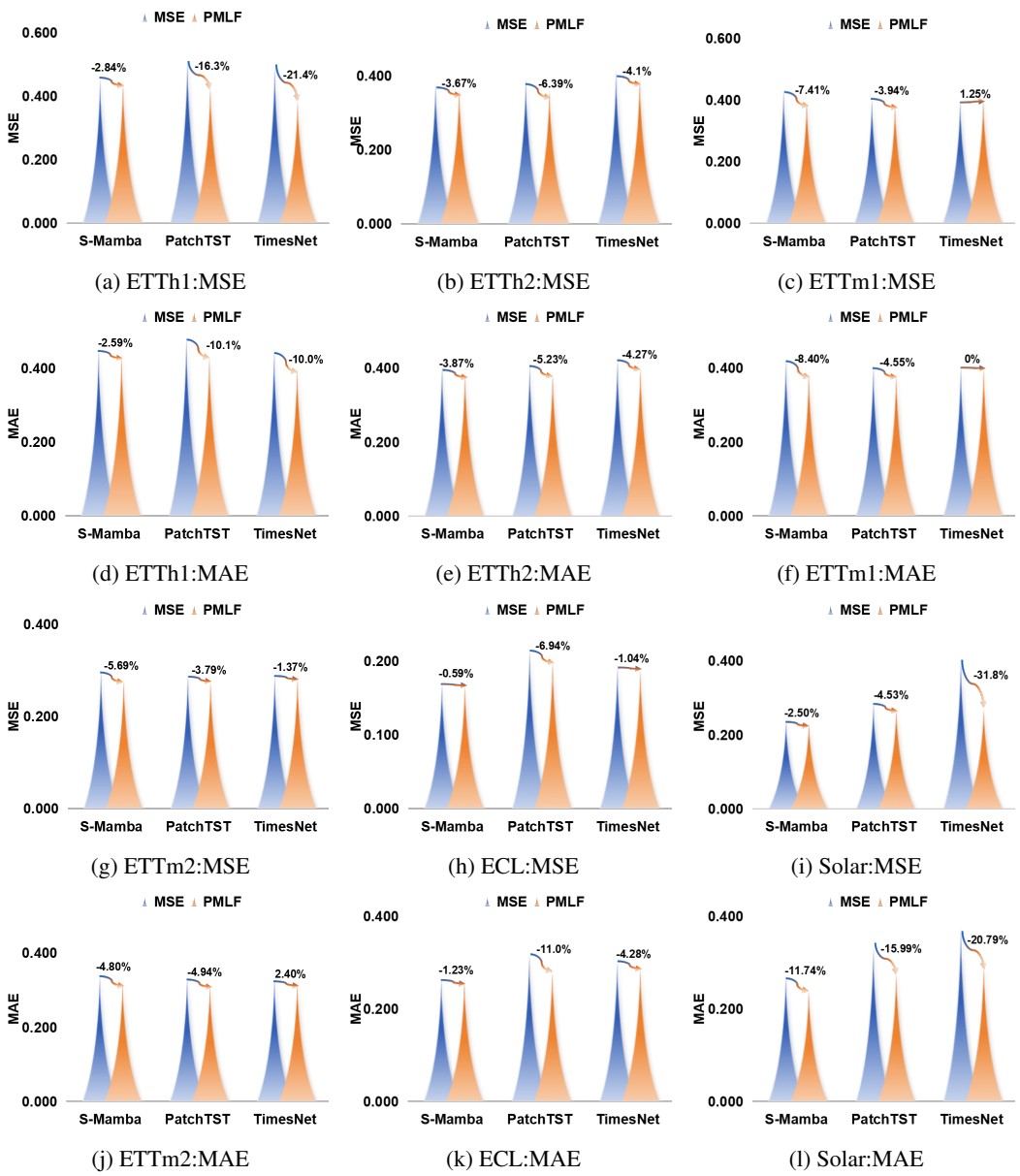

Figure 8: Average forecasting performance of PatchTST, S-Mamba, and TimesNet under four prediction horizons (96, 192, 336, 720) on the ETT (4 subsets), Weather, ECL, and Solar datasets. Models trained with PMLF are contrasted with their original MSE counterparts. Across all horizons and datasets, PMLF consistently reduces MAE, MSE, and RMSE, illustrating its effectiveness on these classical architectures.

generated using PMLF exhibit closer alignment with the ground-truth signals, capturing long-term trends and periodic patterns with higher fidelity compared to those produced with MSE.

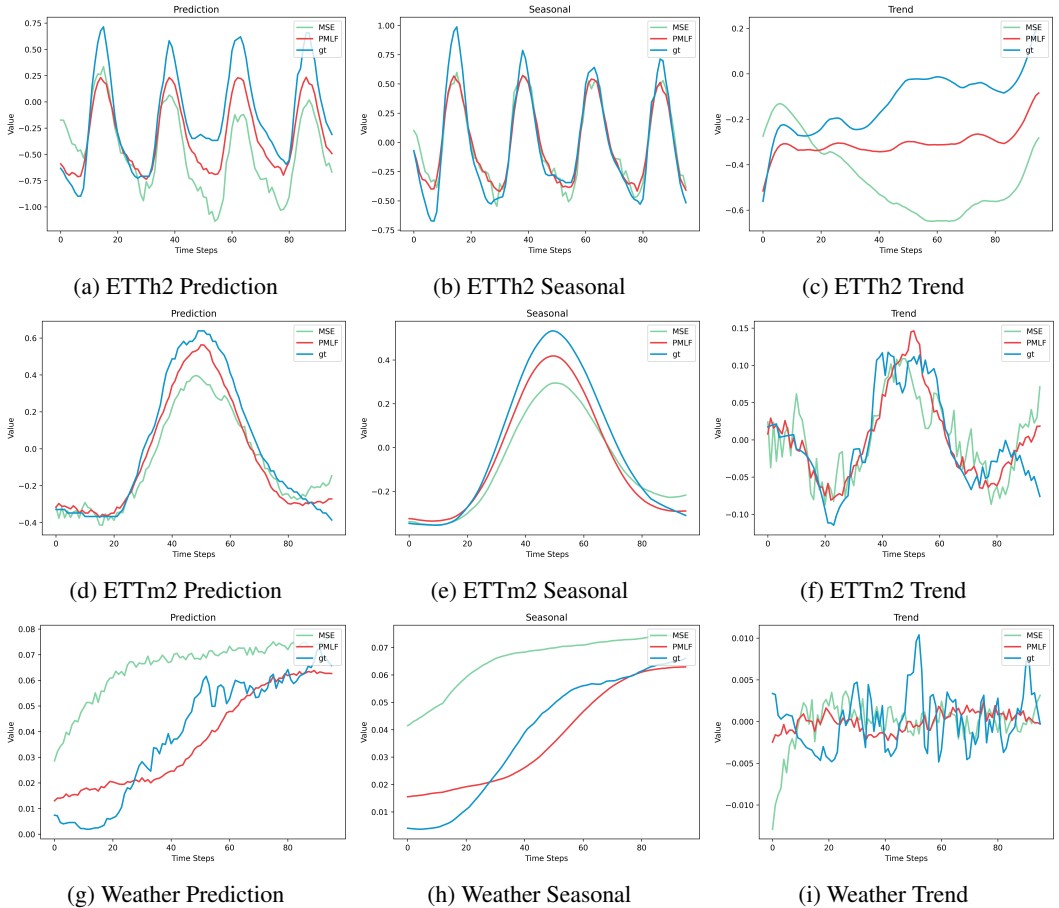

Figure 9: Visual comparison of Amplifier forecasts trained with PMLF and MSE across six benchmark datasets. Each row corresponds to one dataset (ETTh2, ETTm2, Weather), with three columns showing the overall prediction (left), seasonal component (middle), and trend component (right).

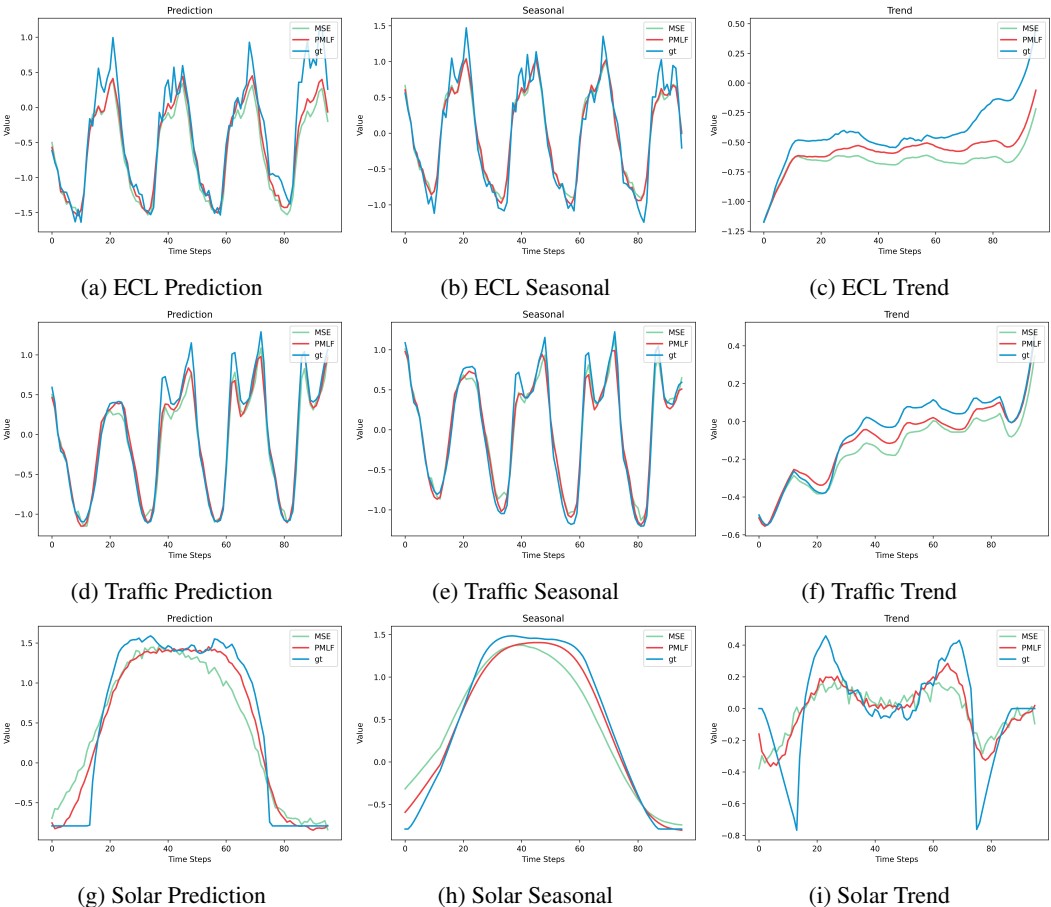

Figure 10: Visual comparison of Amplifier forecasts trained with PMLF and MSE across six benchmark datasets. Each row corresponds to one dataset (ECL, Traffic, and Solar), with three columns showing the overall prediction (left), seasonal component (middle), and trend component (right).

