# OpenReview forum: "PMLF: A Physics-Guided Multiscale Loss Framework for Structurally Heterogeneous Time Series"
_NeurIPS.cc/2025/Conference — NeurIPS 2025 poster_

### Official Review · Reviewer_jE1W · 2025-06-04

**Clarity:** 2
**Significance:** 2
**Originality:** 2
**Rating:** 3
**Confidence:** 5

**Summary:**

This paper introduces a Physics-guided Multiscale Loss Framework (PMLF) for time series forecasting, targeting the structural differences between high-frequency seasonal fluctuations and low-frequency trend dynamics. Unlike existing methods that use a unified loss, PMLF decomposes time series into seasonal and trend components, applying a quadratic loss to the former and a logarithmic loss to the latter. A softmax-based weighting mechanism dynamically balances their contributions. By aligning supervision with the underlying physical behavior of components, PMLF improves forecasting performance across multiple benchmarks, effectively addressing structural heterogeneity in multiscale time series data.

**Questions:**

Please refer to the weakness.

**Ethical Concerns:**

["NO or VERY MINOR ethics concerns only"]

**Limitations:**

yes

**Quality:**

2

**Strengths And Weaknesses:**

Strengths:
1. The paper introduces a physics-inspired, component-specific loss formulation that aligns well with the distinct behaviors of seasonal and trend components, offering a novel perspective on supervision in time series forecasting.

2. PMLF can be seamlessly applied across diverse backbone architectures, demonstrating strong generalization and consistent performance improvements on multiple benchmarks.


Weaknesses:
1. The definitions of s(t) and \tau are not provided. Please clarify their relationship to Y^s  and Y^t, respectively.

2. Equation (7), labeled as the “seasonal loss,” appears mathematically equivalent to the standard MSE applied to the seasonal component. This raises concerns about whether the proposed energy-based loss introduces a genuinely new formulation or merely a reinterpretation of MSE through a physical analogy. The manuscript should clarify whether the energy-based perspective yields any theoretical or empirical advantage beyond reinterpretation. Additionally, it remains unclear how this formulation addresses phase or amplitude shifts. Including a concrete example could significantly aid comprehension.

3. The relaxation energy loss, expressed via a logarithmic penalty (Equation 10), seems to introduce robustness to outliers due to its dampened gradient for large deviations. However, the manuscript lacks a clear explanation of its practical implication. A discussion on its theoretical advantage and empirical behavior in presence of temporal drift or outliers would be beneficial.

4. Since DTW is designed to handle phase shift, a comparison or discussion is warranted. What are the advantages of the proposed loss functions relative to DTW?

5. The trend prediction shown in Figure 1 exhibits unexpected oscillatory behavior. This is counterintuitive given the assumed smoothness of trend components. Providing quantitative evidence or theoretical justification for this phenomenon would strengthen the paper’s claims. Furthermore, the concept of "temporal drift" requires clearer definition. How is it distinct from standard deviation?

6. The paper currently lacks a comprehensive comparison against structure-aware baselines across all seven datasets. Including such experiments would provide a more convincing argument for the generalizability and effectiveness of the proposed method.

---

> ### Author Rebuttal · Authors · 2025-07-31
>
> **[W1] The definitions of s(t) and \tau and their relationship to Y^s and Y^t, respectively.**
>
> **Response.** In our framework, the multi variate time series $Y(t) \in \mathcal{R}^{C \times T}$ is decomposed into two components via seasonal-trend decomposition:
>
> $Y=Y_s+Y_\tau$
>
> where $Y_s$ represents the seasonal component and $Y_\tau$ denotes the trend component. For notational clarity, **$s(t)$ and $\tau(t)$ respectively refer to one individual channel of the seasonal $Y_s$ and trend components $Y_\tau$**. We sincerely apologize for not explaining this more clearly in the original manuscript, and will revise the notation and provide explicit definitions in the updated version to ensure better readability.
>
> ---
>
> **[W2] concerns about whether the proposed energy-based loss introduces a genuinely new formulation or merely a reinterpretation of MSE through a physical analogy. how this formulation addresses phase or amplitude shifts.**
>
>
> **Response.** Thank you for raising this critical and insightful question. We would like to clarify that although Equation (7) takes the form of a quadratic penalty, it is not a heuristic reinterpretation of MSE. It is derived from Potential energy of harmonic oscillator：$U_{vib}(x)=k/2(x-x_0)^2$ and the seasonal loss is defined as the difference of potential energies $\Delta U_{vib}$.
>
> **Empirical Validation.** To verify whether this physically inspired design yields real benefits, we compared several alternative combinations of loss functions for seasonal/trend components:Quadratic + Quadratic (Q+Q), Logarithmic + Logarithmic (L+L), Swapped (L+Q) and Heuristic combinations including MSE+MAE, MAE+Huber, and Huber+MSE. Due to character limitations, we report average forecasting results across four prediction lengths (96, 192, 336, 720) using Amplifier as the backbone:
>
> |Loss|PMLF||Q+Q||L+L||L+Q||MSE+MAE||MAE+Huber||Huber+MSE||
> |---|---|---|---|---|---|---|---|---|---|---|---|---|-|-|
> |Metric|MSE|MAE|MSE|MAE|MSE|MAE|MSE|MAE|MSE|MAE|MSE|MAE|MSE|MAE|
> |ETTh1|**0.443**|**0.434**|0.480|0.453|0.455|0.436|0.481|0.451|0.453|0.439|0.462|0.440|0.480|0.453|
> |ETTh2|**0.356**|**0.385**|0.377|0.404|0.363|0.387|0.387|0.386|0.366|0.390|0.369|0.396|0.383|0.408|
> |ETTm1|0.391|**0.386**|0.390|0.399|0.396|0.387|0.389|0.398|0.392|0.390|0.387|0.393|0.391|0.400|
> |ETTm2|0.275|0.316|0.313|0.347|0.277|0.322|0.279|0.325|0.275|0.317|**0.274**|0.316|0.281|**0.252**|
> |Weather|**0.242**|**0.263**|0.248|0.274|0.245|0.268|0.248|0.274|0.243|0.264|0.244|0.270|0.245|0.274|
>
> These results suggest that PMLF is not only theoretically motivated but consistently outperforms all other configurations, including heuristic and purely statistical losses.
>
> **Sensitivity to Phase and Amplitude Shifts.** Seasonal losses mainly come from two types: amplitude shift and phase shift.
> - **Amplitude Shift.** A baseline offset in amplitude causes consistent deviations over time, leading to cumulative penalties. The larger the offset, the stronger the loss, encouraging the model to align the predicted baseline with the ground truth.
>
> - **Phase Shift.** Even slight phase misalignments can significantly amplify the structural loss, as the predicted peaks and troughs become temporally misaligned with the ground truth. This sharp sensitivity to phase errors causes the loss to escalate rapidly, encouraging the model to correct temporal misalignment even when the overall waveform shape is preserved.
>
> To illustrate this, we use a sine function $y=sin(2\pi x)$, and compute mse loss variations under controlled shifts:
>
> |$\Delta$ amplitude |$\Delta loss_a$||$\Delta$ phase|$\Delta loss_p$|
> |---|---|---|---|---|
> |0.1|0.005||$5^{\circ}$|0.004
> |0.2|0.02||$10^{\circ}$|0.015
> |0.5|0.125||$30^{\circ}$|0.132
>
> These results confirm that even slight amplitude or phase deviations lead to noticeable increases in loss, demonstrating that our formulation effectively penalizes these structural mismatches.
>
> ---
>
> **[W3] Clarification on the  theoretical advantage and empirical behavior of the relaxation energy loss**
>
> **Response.** Thank you for your professional and constructive feedback. While the manuscript outlines the relaxation energy loss design, we agree its robustness implications deserve further clarification.
>
>
> - **Theoretical Advantage.** The relaxation loss adopts a logarithmic form $\log(1 + | \hat{\tau} - \tau |)$, which imposes moderate penalities on large deviations. Unlike MSE or MAE, this design yields smoother gradients and enhances robustness against outliers and temporal drift, which is critical for stable long-term trend modeling.
> - **Empirical behavior.** The real-world datasets used in our study naturally exhibit temporal drift and contain substantial outliers due to noise and irregular trends. We compare the relaxation loss with commonly used robust losses such as MAE and Huber. As shown below, our method consistently achieves superior performance under such noisy conditions, validating its effectiveness in suppressing the influence of anomalous deviations while preserving structural trend signals.
>
> |Dataset|PMLF(MSE/MAE)|Huber(MSE/MAE)|MAE(MSE/MAE)
> |-|-|-|-
> |ETTh1|**0.443**/**0.434**|0.475/0.445|0.459/0.447
> |ETTh2|**0.356**/**0.385**|0.372/0.399|0.369/0.394
> |ETTm1|0.391/**0.386**|**0.388**/0.392|0.394/0.386
> |ETTm2|**0.275**/**0.316**|0.277/0.322|0.276/0.318
> |Weather|**0.242**/**0.263**|0.244/0.268|0.243/0.263
>
> ---
>
> **[W4] What are the advantages of the proposed loss functions relative to DTW**
>
> **Response.** Thanks your professional and insightful opinion. Introducing more loss functions, including DTW, for comparison will further demonstrate the advantages of our loss function design. we provide a comprehensive comparison highlighting PMLF’s theoretical and empirical advantages.
>
> - **Differentiability** DTW is inherently non-differentiable and thus unsuitable as a loss for gradient-based optimization. Although Soft-DTW enables differentiability via smoothing, it introduces approximation bias. In contrast, PMLF is fully differentiable, enabling precise and stable gradient updates for both seasonal and trend components.
> - **Efficiency.** Soft-DTW has $O(T^2)$ time and memory complexity, which scales poorly for long sequences. PMLF operates in linear time $O(T)$, enabling fast training and deployment.
> - **Structural Decomposition.** DTW aligns sequences holistically based on overall shape but lacks the ability to disentangle different structural sources of error. PMLF explicitly decomposes the prediction into seasonal and trend components, allowing the model to receive targeted supervision for each structural mode.
> - **Loss Effectiveness.** In experiments across five datasets （below table）, PMLF consistently achieves lower MSE and MAE compared to Soft-DTW, demonstrating its superior effectiveness in guiding model training.
>
> |Dataset|**PMLF (MSE/MAE)**|Soft-DTW (MSE/MAE)|
> |---|---|---|
> |ETTh1|**0.443 / 0.434**|0.463 / 0.443|
> |ETTh2|**0.356 / 0.385**| 0.378 / 0.405|
> |ETTm1|0.391 / **0.386**| **0.389** / 0.399|
> |ETTm2|**0.275 / 0.316**| 0.282 / 0.328|
> |Weather|**0.242 / 0.263**| 0.245 / 0.273|
>
> **[W5]Clarification on oscillatory trend prediction and definition of temporal drift.**
>
> **Response.**
> -  **Oscillatory Trend Prediction.** The mild oscillations observed in the predicted trend result from the scale-dependent nature of trend and seasonality decomposition. Patterns that appear as trends over short time windows, such as hours or days, may correspond to low-frequency seasonal variations when viewed across longer time frames. This ambiguity is common in real-world time series. Our relaxation loss could accommodate gradual structural drift while suppressing rapid fluctuations, allowing the model to capture stable yet realistic trend dynamics without excessive smoothing.
>
> - **Definition of Temporal Drift.** In time series, temporal drift refers to the gradual, cumulative shift in the baseline level of a time series over time. It is distinct from standard deviation, which measures short-term variability around the local mean. To illustrate this, we refer to the well-known AirPassengers dataset (recorded monthly). As shown below, while the within-year variance (±std) remains relatively stable, the month;y mean increases significantly from 126.67 in 1949 to 197.00 in 1952:
>
> |year|1949|1950|1951|1952
> |-|-|-|-|-
> |mean±std|126.67±13.72|139.67±19.07|170.17±18.44|197.00±22.97
> ---
> **[W6] The paper currently lacks a comprehensive comparison against structure-aware baselines across all seven datasets. Including such experiments would provide a more convincing argument for the generalizability and effectiveness of the proposed method**
>
> **Response.**  Thank you for your comprehensive and professional advice. We have expanded the comparison of structural losses to seven datasets. The following is a comparison of the mean values of their four different prediction lengths:
>
> | Loss|PMLF||FreDF||TIDE-Q||
> |---|---|---|---|---|---|---|
> |Metric|MSE|MAE|MSE|MAE|MSE|MAE|
> |ETTh1|**0.443**|**0.434**|0.450|0.435|0.478|0.449||
> |ETTh2|**0.356**|**0.385**|0.364|0.391|0.366|0.394|
> |ETTm1|**0.391**|**0.386**|0.393|0.400|**0.391**|0.395|
> |ETTm2|0.275|**0.316**|**0.273**|0.317|0.275|0.318|
> |Weather|**0.242**|**0.263**|0.248|0.273|0.249|0.274|
> |ECL|**0.170**|**0.259**|**0.170**|0.262|0.176|0.270|
> |Traffic|**0.475**|**0.310**|0.481|0.303|0.482|0.316|
>
> The above results indicate that although FreDF and TIDE-Q loss have some competitiveness on a very small number of datasets, our proposed PMLF is applicable to all datasets, which demonstrates the universality and effectiveness of our method.
>
> ---
>
> We appreciate the reviewers’ thoughtful comments. In response, we have carefully examined and addressed all aspects of loss design, ablation comparison, conceptual grounding, and training behavior, supported by experiments and analysis. The final version will include additional resources to improve reproducibility and understanding.

---

> ### Comment · Reviewer_jE1W · 2025-08-05
>
> W2. Despite the seemingly rich physical meaning of potential energy, the strong assumption given by Equation 6 reduces the original functional form to a simple MSE. I therefore still hold the view that this is merely a reinterpretation.
>
> W3. The textual explanation for the relaxation energy loss's robustness is not sufficient. I recommend the authors provide a mathematical justification instead of a purely textual one to enhance the rigor of their clarification.
>
> W5 Similar to W3, oscillatory behavior and temporal drift require a mathematical definition.
>
> I appreciate the authors' efforts in responding to my questions. However, given these unresolved issues and the other acknowledged limitations, I believe my initial rating is a fair evaluation of this manuscript and will maintain it.

---

> > ### Author Response · Authors · 2025-08-06
> >
> > **[W2] Response.** We appreciate the reviewer’s concern. We respectfully clarify that Equation 6 is not a mere reinterpretation of MSE, but a physically and statistically justified simplification derived from the behavior of periodic signals. For an idealized harmonic oscillator, the seasonal component $s(t)$ satisfies:
> > $$
> > s(t) = A \sin(\omega t + \phi), \quad \mathbb{E}[s(t)] = 0.
> > $$
> >
> > Given a prediction $\hat{s}(t)$ with error $\delta s(t) = \hat{s}(t) - s(t)$, the cross-term in the energy function becomes:
> > $$
> > \mathbb{E}[(s(t) - x_0)(\hat{s}(t) - s(t))] = \mathbb{E}[s(t)\delta s(t)] - x_0 \cdot \mathbb{E}[\delta s(t)].
> > $$
> > This term vanishes as $x_0 = \mathbb{E}[s(t)] = 0$, $\delta s(t)$ is asymptotically uncorrelated with $s(t)$, and averaging is over complete or stationary periods. Hence,
> > $$
> > \mathbb{E}[(s(t) - x_0)(\hat{s}(t) - s(t))] \approx 0.
> > $$
> >
> > The approximation holds naturally over complete oscillatory periods, without the need for any strong assumptions. It remains valid regardless of amplitude $A$ or phase $\phi$, and aligns with the underlying physics of stable oscillatory processes.
> >
> > To further validate this formulation, we **reintroduce the full potential energy expression (including the cross term)** and evaluate it empirically. The results below demonstrate that this variant still consistently outperforms MSE:
> >
> > Dataset| PMLF||PMLF+Cross Term|| MSE||
> > -|-|-|-|-|-|-|
> > Metric|MSE|MAE|MSE|MAE|MSE|MAE
> > ETTh2|0.356|0.385|0.364|0.387|0.383|0.409
> > ETTm2|0.275|0.316|0.277|0.317|0.281|0.327|
> > Weather|0.242|0.263|0.243|0.264|0.247|0.274
> > Traffic|0.475|0.310|0.479|0.316|0.482|0.315
> >
> > These empirical results reinforce the soundness of our physical approximation and confirm that the proposed design is not a reinterpretation of MSE, but a meaningful, principled, and effective formulation.
> >
> > ---
> >
> > **[W3] Response.**
> > We appreciate the reviewer’s suggestion. To substantiate the robustness of relaxation loss, we present a formal sensitivity analysis. The relaxation loss is defined as:
> > $$
> > L_{\tau}(e) = \log(1 + \alpha |e|), \quad e = \hat{\tau} - \tau.
> > $$
> > with gradient and curvature:
> > $$
> > \nabla L_{\tau} = \frac{\alpha\, \mathrm{sgn}(e)}{1 + \alpha |e|}, \quad
> > \nabla^2 L_{\tau} = -\frac{\alpha^2}{(1 + \alpha |e|)^2}.
> > $$
> >
> > This formulation exhibits the following desirable properties:
> >
> > (i) **Bounded gradient**: $\nabla L_{\tau} \to \alpha$ as $|e| \to \infty$, which limits the influence of large deviations;
> >
> >  (ii) **Vanishing curvature**:  $\nabla^2 L_{\tau} \to 0$ as $|e| \to \infty$, reducing the penalization on outliers.
> >
> > Compared with MSE and MAE, relaxation loss provides smoother optimization and greater resilience to noise. A summary table is shown below:
> > |Loss  | $\nabla$ upper bound | $\nabla^2$ upper bound | Sensitivity to outliers|
> > |--|--|--|--|
> > | MSE | $\infty$ | Constant  | High  |
> > | MAE | Constant | 0         | Medium|
> > | Relaxation | $\alpha$ | 0  | Low   |
> >
> > These analytical results align with our empirical findings presented in W3, where the relaxation loss consistently outperforms MSE and MAE.
> >
> > ---
> >
> > **[W5] Response.**  Thank you for the suggestion. To rigorously define the notions of oscillatory behavior and temporal drift, we provide both mathematical formulations and theoretical insights.
> >
> > **Oscillatory Behavior.**  Oscillations in the trend prediction arise from the scale-dependent nature of trend–seasonality decomposition. When seasonal components contain low-frequency energy, they may leak into the trend through imperfect filtering. Formally, if the seasonal term is given by
> > $$
> > s(t) = A \sin(\omega t + \phi),
> > $$
> > and $\omega$ is close to the trend frequency, $s(t)$ will pass through the low-pass filter and manifest as oscillations in the predicted trend $\tau(t)$.
> >
> > **Temporal Drift.**  Temporal drift refers to the cumulative shift in the trend baseline over time. It captures long-term structural changes beyond short-term fluctuations. Mathematically, it is defined as:
> > $$
> > \Delta \tau(t) = \tau(t) - \tau(t_0),
> > $$
> > where $\tau(t)$ is the trend component and $t_0$ is the initial time. This formulation distinguishes long-term drift from standard deviation, which reflects local variability.
> >
> > We provide a clear mathematical explanation for the oscillatory behavior in trend predictions and define temporal drift rigorously. The oscillations are due to low-frequency seasonal components leaking into the trend, while temporal drift captures long-term cumulative changes distinct from short-term variability. These insights are supported by both theoretical analysis and empirical evidence.

---

> > ### Author Response · Authors · 2025-08-08
> >
> > Dear Reviewer jE1W,
> >
> > As the discussion phase is approaching its end, we would like to gently remind you of our earlier response to your concern. Your feedback is very important to us, and we sincerely appreciate the time and thought you have dedicated to reviewing our work.
> >
> > Thank you again for your time and consideration.
> >
> > Best regards,
> > 7051 Authors

---

### Official Review · Reviewer_mT8C · 2025-06-25

**Clarity:** 3
**Significance:** 3
**Originality:** 3
**Rating:** 5
**Confidence:** 3

**Summary:**

This paper proposes a Physics-guided Multiscale Loss Framework (PMLF) that decomposes time series into seasonal and trend components, then designs physics-inspired loss functions for each (quadratic loss for seasonal components mimicking molecular vibration energy, logarithmic loss for trend components simulating molecular drift relaxation energy). By incorporating dynamic weighting to balance their optimization, PMLF aims to address the limitations of traditional uniform loss functions in simultaneously capturing short-term fluctuations and long-term trends. Experimental results demonstrate PMLF's ability to improve prediction accuracy across multiple baseline models.

**Questions:**

1. Could you provide formal analysis on the convergence properties of your proposed loss functions? Specifically, under what conditions does the dynamic weighting mechanism guarantee balanced optimization between seasonal and trend components?

**Ethical Concerns:**

["NO or VERY MINOR ethics concerns only"]

**Final Justification:**

The revisions have addressed my concerns regarding the details. Based on my original positive score, I will maintain my score.

**Quality:**

3

**Strengths And Weaknesses:**

Strengths:
1.The paper provides a well-motivated theoretical foundation, drawing clear analogies between molecular dynamics and time series components. The loss functions are derived from physical principles, and the dynamic weighting mechanism is carefully justified

2.Extensive experiments across multiple datasets and diverse baselines demonstrate consistent improvements in both MSE and MAE.

Weaknesses:
1. The role of the balancing parameter (β) is briefly discussed, but its impact on different datasets/models could be analyzed further.

2. computational complexity is not adequately addressed, as the decomposition operations and dynamic weighting mechanism may introduce non-negligible overhead.

---

> ### Author Rebuttal · Authors · 2025-07-31
>
> Thank you very much for your positive evaluation and appreciation of our universality, and empirical performance. Here are our responses to specific questions and concerns.
>
> ### **[Q]: Could you provide formal analysis on the convergence properties of your proposed loss functions? Specifically, under what conditions does the dynamic weighting mechanism guarantee balanced optimization between seasonal and trend components?**
>
> **Response.** We appreciate the reviewer’s insightful question. We address this from two perspectives: (1) the convergence behavior of the proposed loss functions, and (2) the conditions under which the dynamic weighting mechanism ensures balanced optimization between seasonal and trend components.
>
> - **Convergence Properties.** Our total loss is defined as:
>
> $$
> \mathcal{L}\_{\text{total}} = \lambda\_s \mathcal{L}_s + \lambda\_{\tau} \mathcal{L}\_{\tau}
> $$
> The seasonal loss $\mathcal{L}_s$ is defined as a convex quadratic penalty:
>
> $$
> \mathcal{L}\_s = \frac{1}{CH} \sum\_{c=1}^C \sum\_{h=1}^H( \hat{s}^{(c)}\_{T+h} - s^{(c)}\_{T+h})^2
> $$
>
> which ensures stable convergence under gradient-based optimization. The trend loss $\mathcal{L}_\tau$ adopts a logarithmic form:
>
> $$
> \mathcal{L}\_\tau = \frac{1}{CH} \sum\_{c=1}^C\sum\_{h=1}^H \log(1 + | \hat{\tau}^{(c)}\_{T+h} - \tau^{(c)}\_{T+h} | )
> $$
>
> with gradient:
>
> $$
> \frac{d \mathcal{L}_{\tau}}{de} = \frac{1}{1 + e} \in (0,1]
> $$
>
> Although $\mathcal{L}\tau$ is non-convex, it is Lipschitz-continuous and differentiable, ensuring that the total loss $\mathcal{L}{\text{total}}$ remains smooth. Empirically, we observe consistent monotonic decrease of $\mathcal{L}_{\text{total}}$ across all tasks, validating its practical convergence behavior. Due to submission restrictions, we will supplement the convergence curve in the future.
>
> - **Balanced Optimization via Dynamic Weighting.** To ensure balanced optimization between the seasonal and trend components, we design a softmax-based dynamic weighting strategy:
>
> $$ \lambda\_s = \frac{\exp(\beta (\mathcal{L}\_s^\dagger - m))}{\exp(\beta (\mathcal{L}\_s^\dagger - m)) + \exp(\beta (\mathcal{L}\_\tau^\dagger - m))}, \quad
>     \lambda\_\tau = \frac{\exp(\beta (\mathcal{L}\_\tau^\dagger - m))}{\exp(\beta (\mathcal{L}\_s^\dagger - m)) + \exp(\beta (\mathcal{L}\_\tau^\dagger - m))} $$
>
> where $\mathcal{L}s^\dagger$ and $\mathcal{L}\tau^\dagger$ denote the detached seasonal and trend losses (no gradient flow),$m = \max(\mathcal{L}s^\dagger, \mathcal{L}\tau^\dagger)$ ensures numerical stability. This mechanism possesses several important properties:
>
> - **Responsiveness to imbalance.** If one component (e.g., trend) exhibits significantly larger loss, its corresponding weight $\lambda_\tau$ will increase, prompting the model to focus more on the underperforming part. This behavior aligns with the principle of adaptive error compensation.
> - **Stability through max-normalization.** The use of $m = \max(\cdot)$ shifts both logits before exponentiation, avoiding disproportionate weight assignment. For example, if one loss is much larger than the other, direct softmax could lead to almost binary weights (e.g., [1, 0]); the max-shifting avoids this by anchoring the maximum at zero and reducing the gap between logits, thus ensuring neither component is neglected.
> - **Temperature control with $\beta$.** The sharpness of this rebalancing is governed by $\beta$: A higher $\beta$ emphasizes the relative difference between the two losses, leading to sharper focus on the harder task. A lower $\beta$ produces smoother balancing, allowing more gradual joint optimization.
>
> In our experiments, we set $\beta=1$ for stability, but as shown in Figure 5, the results remain robust within a wide range of $\beta \in [0.2, 5]$, highlighting the insensitivity of the mechanism to this hyperparameter.
>
> - **Empirical Evidence.** We further monitor the evolution of dynamic weights across four prediction lengths on ETTh1 using Amplifier as the backbone. Although minor fluctuations are observed in the early iterations due to initial loss disparity, the weights consistently converge to a stable state within a few dozen steps. The table below reports the mean±std after convergence, confirming that the dynamic weighting mechanism promotes a balanced optimization between seasonal and trend losses without oscillation or dominance.
>
> |Horizon|Seasonal Weight| Trend Weight|
> |-|-|-|
> |96 | 0.4810 ± 0.0039| 0.5190 ± 0.0039|
> |192| 0.4779 ± 0.0040| 0.5221 ± 0.0040|
> |336| 0.4854 ± 0.0037| 0.5146 ± 0.0037|
> |720| 0.4679 ± 0.0045| 0.5321 ± 0.0045|
>
> ---
>
> ### **[W1] The role of the balancing parameter (β) impact on different datasets/models.**
>
> **Response.** We thank the reviewer for the insightful question regarding the balancing coefficient $\beta$, which controls the relative weight between the seasonal and trend loss components. To assess its impact, we conduct controlled experiments across three backbone models (TimeMixer, Amplifier, TimeXer) and two benchmark datasets (ETTh2 and Weather). Results are summarized below.
>
> Performance across different models (ETTh2)
> |method|TimeMixer||Amplifier||TimeXer||
> |----|---|---|---|---|---|---|
> |$\beta$|MSE|MAE|MSE|MAE|MSE|MAE|
> |0.2|0.364|0.389|**0.359**|0.385|**0.361**|0.389|
> |0.4|**0.363**|**0.388**|0.360|**0.384**|0.362|0.389|
> |0.6|**0.363**|**0.388**|**0.359**|**0.384**|0.363|**0.388**|
> |0.8|0.365|0.389|0.362|0.386|0.362|**0.388**|
> |1.0|0.369|0.391|0.363|0.387|0.364|0.389|
> |2.0|0.372|0.392|0.363|0.388|0.365|0.390|
> |3.0|0.370|0.392|0.371|0.393|0.367|0.392|
> |5.0|0.372|0.395|0.374|0.396|0.375|0.398|
>
> Across all models, best performance consistently occurs within $\beta$ ∈ [0.4, 0.8]. When $\beta$ is too large, model performance degrades significantly. This suggests that overly emphasizing one component can impair learning balance and generalization.
>
> Dataset|ETTh2||Weather||
> |-|-|-|-|-
> $\beta$|MSE|MAE|MSE|MAE|
> 0.2|**0.361**|0.389|0.242|0.264|
> 0.4|0.362|0.389|0.242|0.264|
> 0.6|0.363|**0.388**|0.240|0.262|
> 0.8|0.362|**0.388**|0.240|0.262|
> 1.0|0.364|0.389|**0.239**|**0.261**|
> 2.0|0.365|0.390|0.241|0.263|
> 3.0|0.367|0.392|0.241|0.263|
> 5.0|0.375|0.398|0.244|0.267|
>
> On both ETTh2 and Weather, the optimal $β$ also lies between 0.4 and 1.0, with Weather favoring a slightly larger $β$. Notably, performance drops when $β$ becomes too large, consistent with our prior observations.
>
> **Conclusion.** Overall, the balancing coefficient \$\beta\$ performs best within the range of **0.4 to 1.0** across models and datasets. When \$\beta\$ becomes excessively large, model performance consistently deteriorates. This confirms that overly skewing the loss weights disrupts the balance between seasonal and trend learning, leading to suboptimal results.
>
> ---
>
> ### **[W2] computational complexity is not adequately addressed, as the decomposition operations and dynamic weighting mechanism may introduce non-negligible overhead.**
>
> We appreciate the reviewer’s concern regarding potential computational overhead. We would like to clarify that both the decomposition and dynamic weighting mechanism are implemented in a lightweight and efficient manner, **incurring negligible additional cost relative to the backbone network.**
>
> - **computational complexity.** We adopt a simple moving average filter for time series decomposition, with a computational complexity of $O(T)$, where $T$ is the sequence length. This decomposition method is memory-efficient, parallelizable, and incurs minimal time cost in practice.
>
> - **Dynamic Weighting Cost.** The softmax-based dynamic weighting is applied once per batch and involves only two scalar loss $\mathcal{L}\_s$ and $\mathcal{L}\_{\tau}$. The additional computation amounts to two exponentials and one division, making the complexity constant-time $O(1)$ and negligible compared to the deep learning models.
>
> To quantify its practical cost, we report the average runtime introduced by our loss function (including decomposition and weighting) on the ETTh1 dataset with a batch size of 64, under different prediction horizons:
>
> Horizon|Cost Time(ms)|
> -|-|
> 96|0.6186±0.0294
> 192|0.6223±0.0325
> 336|0.6474±0.0234
> 720|0.6599±0.0235
>
> Even for long-horizon forecasts, the additional cost remains below 0.7 ms per batch, which is negligible compared to the total training time.
>
> ---
>
> In summary, we provide both theoretical grounding (Lipschitz continuity, descent lemma) and empirical validation (convergence curves, β-sensitivity, runtime profiling) to guarantee balanced optimization and negligible overhead.  .Additional materials and clarifications will be provided in the final version to further enhance transparency and reproducibility.

---

> > ### Author Response · Authors · 2025-08-06
> >
> > Dear Reviewer mT8C,
> >
> > We sincerely thank you for your time and thoughtful review.
> >
> > We have carefully addressed your comments in our rebuttal. If you happen to have any further thoughts or concerns, we would be more than happy to clarify and provide any additional information that might be helpful.
> >
> > We fully understand your busy schedule, and we truly appreciate your support in improving our work.
> >
> > Many thanks,
> >
> > 7051 Authors

---

### Official Review · Reviewer_eskT · 2025-07-03

**Clarity:** 3
**Significance:** 2
**Originality:** 3
**Rating:** 4
**Confidence:** 3

**Summary:**

This paper proposes PMLF, a novel loss framework tailored for time series with structurally heterogeneous components (i.e., seasonal and trend). Drawing inspiration from physical systems, the authors design two specialized loss functions: a quadratic form for high-frequency seasonal components (analogous to harmonic oscillations), and a logarithmic form for low-frequency trend components (analogous to structural relaxation). A softmax-based dynamic weighting mechanism is further introduced to balance the optimization across these two losses. Empirical results on a wide range of datasets and backbones suggest that PMLF improves over standard MSE losses and structure-aware baselines.

**Questions:**

1. How critical is the choice of quadratic and logarithmic loss functions? Have you evaluated alternative forms (e.g., Huber, MAE, or frequency-weighted losses) to assess the necessity of the proposed physical analogies?
2. How does the dynamic weighting mechanism evolve during training? Can you provide a plot of the trend vs. seasonal weights over epochs, and clarify whether this weighting stabilizes or oscillates?
3. Have you tested the learnable decomposition strategy in a controlled way? It’s briefly mentioned but not quantitatively compared with the fixed-kernel approach. Does it improve performance on nonstationary or high-frequency data?
4. Can PMLF be extended to handle more than two components? For time series that include noise, regime shifts, or multiple periodicities, can the loss framework be generalized to supervise more than just trend and seasonal parts?
5. What are the computational implications of the proposed loss? Does the log-based trend loss introduce optimization difficulties (e.g., vanishing gradients), or slow down convergence compared to MSE?

**Ethical Concerns:**

["NO or VERY MINOR ethics concerns only"]

**Final Justification:**

Basically, most questions are answered in the rebuttal, while some points may go beyond this current work. I would like to keep my current rating.

**Limitations:**

1. The effectiveness of PMLF relies on the assumption that the input time series can be cleanly decomposed into distinguishable trend and seasonal components. In real-world applications such as medical signals, high-frequency financial data, or irregular event streams, this assumption may not hold, limiting the applicability of the method.

2. The current design focuses exclusively on a two-part decomposition (trend and seasonal). It is unclear whether the framework can be extended to time series with more complex or hierarchical structures (e.g., noise, abrupt shifts, multi-scale events).

**Paper Formatting Concerns:**

n.a.

**Quality:**

3

**Strengths And Weaknesses:**

Strengths
1. The paper rightly identifies a limitation in current time series forecasting models that apply a uniform loss across structurally different components.
2. The physical analogies provide an interesting perspective and yield two loss forms that align well with different structural sensitivities in time series.
3. PMLF is model-agnostic and integrates cleanly into a wide variety of forecasting architectures, as shown in the experiments.
4. The method demonstrates performance improvements across several benchmarks and baselines, and includes ablation studies and robustness analysis.
Weaknesses
1. The proposed use of quadratic and logarithmic losses is based on analogies to molecular dynamics (vibrations and structural drift). While intuitive, this physical metaphor lacks formal statistical or optimization-theoretic grounding. There is no analysis comparing the proposed loss forms to standard alternatives (e.g., MAE, Huber, or frequency-aware metrics), nor a discussion on why these loss functions are optimal or necessary for the intended structural components.
2. The softmax-based weighting scheme introduces a dynamic balance between trend and seasonal losses during training. However, the paper does not analyze how this mechanism behaves over time, whether it converges, or if it may cause instability due to loss scale mismatch. The lack of gradient norm or loss trajectory analysis limits confidence in its robustness.
3. The trend-seasonal separation relies on fixed-kernel moving average filters, which may not be sufficient for data with nonstationary, overlapping, or nonlinear structures. Although a learnable decomposition is mentioned, it is not systematically evaluated or compared, leaving open the question of how critical the decomposition quality is to overall performance.

---

> ### Author Rebuttal · Authors · 2025-07-31
>
> ### **[Q1]How critical is the choice of quadratic and logarithmic loss functions? Have you evaluated alternative forms (e.g., Huber, MAE, or frequency-weighted losses) to assess the necessity of the proposed physical analogies?**
>
> **Response.** Thank you for the thoughtful question. We address the reviewer’s concern from two perspectives: (1) the empirical necessity of using the proposed physically-motivated losses, and (2) the theoretical motivation for pairing quadratic and logarithmic forms in a structured way.
>
> Our design of PMLF stems from the assumption that seasonal and trend components exhibit distinct error characteristics:
> - Seasonal parts behave like bounded oscillations, where symmetric deviations around the mean are natural—thus, quadratic penalty (convex and sensitive to amplitude) is appropriate.
> - Trend parts correspond to slow structural drifts—susceptible to outliers but insensitive to local noise—where logarithmic loss ensures smoother convergence and robustness to scale shifts.
>
> To assess whether our proposed PMLF design is indispensable or replaceable by alternative loss functions, we conducted two classes of ablation:
>  - **Unified Loss Baselines:** MSE, MAE, Huber, and DTW Applied directly to the time series.
>  - **Component-wise Structured Losses:** Pairings applied to seasonal/trend components, including: Q+Q (Quadratic for both components), L+L (Logarithmic for both) and MSE+MAE, Huber+MSE: Non-physics-aligned pairings.
>
> We report the average results across four prediction lengths on five benchmark datasets using the Amplifier backbone. Due to space limits, we show average results of four horizons below:
>
> | Dataset |ETTh1| ETTh2| ETTm1| ETTm2| Weather |
> |-|-|-|-|-|-|
> |Loss Combination|MSE/MAE|MSE/MAE|MSE/MAE|MSE/MAE|MSE/MAE|
> | **PMLF (Q+L)**|**0.443 / 0.434**|**0.356 / 0.385**| 0.391 / **0.386**|**0.275 / 0.316** | **0.242 / 0.263** |
> |Q+Q|0.480 / 0.453|0.377 / 0.404| 0.390 / 0.399| 0.313 / 0.347| 0.248 / 0.274|
> |L+L|0.455 / 0.436| 0.363 / 0.387| 0.396 / 0.387| 0.277 / 0.322| 0.245 / 0.268|
> |MSE+MAE|0.453 / 0.439|0.366 / 0.390|0.392 / 0.390|**0.275** / 0.317|0.243 / 0.264|
> |Huber+MSE|0.480 / 0.453|0.383 / 0.408|0.391 / 0.400|0.281 / 0.327 |0.248 / 0.274|
> Unified MSE|0.478 / 0.451|0.383 / 0.409|**0.384** / 0.398|0.281 / 0.327|0.247 / 0.274|
> Unified MAE|0.459 / 0.447|0.369 / 0.394|0.394 / **0.386**|0.276 / 0.318|0.243 / **0.263**|
> Unified Huber|0.475 / 0.445|0.372 / 0.399|0.388 / 0.392|0.277 / 0.322|0.244 / 0.268
> |Unified DTW|0.463 / 0.443|0.378 / 0.405|0.389 / 0.399|0.282 / 0.328|0.245 / 0.273|
>
> Our results show that:
> - PMLF consistently ranks first or second across all datasets and metrics, outperforming both standard losses and structured pairings.
> - While certain combinations like MSE+MAE achieve competitive results on specific datasets (e.g., ETTm2), they fail to generalize broadly.
> - Unified baselines such as DTW or Huber underperform compared to component-wise modeling, confirming the benefit of decomposition.
> - Symmetric pairings (Q+Q, L+L) lack the asymmetry needed to capture seasonal vs. trend divergence and often suffer from training instability.
>
> We conclude that the proposed Q+L formulation is not only empirically effective but also grounded in meaningful physical structure. The consistent generalization across settings underscores its necessity beyond heuristic loss combinations. We hope this analysis addresses the reviewer’s concerns and demonstrates the necessity of our approach.
>
> ---
>
> ### **[Q2]:How does the dynamic weighting mechanism evolve during training? Can you provide a plot of the trend vs. seasonal weights over epochs, and clarify whether this weighting stabilizes or oscillates?**
>
> **Response.** Thank you for your insightful question. The stability of the dynamic weighting mechanism is crucial to ensuring balanced learning between seasonal and trend components.
>
> While submission constraints prevent us from including figures, we provide the following numerical evidence to support the convergence behavior. We monitored the evolution of seasonal and trend weights across four prediction lengths on ETTh1 using the Amplifier backbone. In all cases, the weights converge within a few dozen iterations. Once stabilized, the weights exhibit low variance and do not oscillate unpredictably throughout training. The table below summarizes the **mean ± standard deviation** of the weights after stabilization:
>
> |Horizon|Seasonal Weight| Trend Weight|
> |-|-|-|
> |96 | 0.4810 ± 0.0039| 0.5190 ± 0.0039|
> |192| 0.4779 ± 0.0040| 0.5221 ± 0.0040|
> |336| 0.4854 ± 0.0037| 0.5146 ± 0.0037|
> |720| 0.4679 ± 0.0045| 0.5321 ± 0.0045|
>
> The low standard deviations (<0.005) across all horizons demonstrate that the learned weights remain highly stable once convergence is reached. These results suggest that the model learns to assign interpretable and horizon-aware importance to seasonal and trend components in a consistent and non-oscillatory manner. We will include full training curves in the final version to visualize the smooth convergence process. We hope this addresses your concern.
>
> ---
>
> ### **[Q3]Effectiveness of learnable decomposition compared to fixed-kernel smoothing.**
>
> **Response.** Thank you for raising this insightful question. We fully agree that a controlled and quantitative comparison is necessary to assess the value of our learnable decomposition strategy, particularly against traditional fixed-kernel smoothing.
>
> We conducted such a comparison in Table 3 (last column: “w/ LMA”) of the main paper. However, we understand it might have been overlooked due to its compact format. To clarify, we extract the relevant entries below, highlighting the difference between:
> - **PMLF:** Our proposed model using fixed-kernel decomposition (smoothed trend).
> - **PMLF with LMA**: A variant that replaces the smoothing operation with a learnable mask aggregation (LMA), allowing data-driven decomposition under the same training configuration.
>
> These results are measured across three representative datasets:
>
> |Dataset|Horizon|PMLF|PMLF with LMA|
> |-|-|-|-|
> |ETTh2|96|**0.283/0.332**|0.289/**0.332**|
> ||192|**0.353/0.379**|0.357/0.379|
> ||336|**0.385/0.405**|0.391/0.406|
> ||720|**0.403/0.423**|0.406/0.425|
> |ETTm2|96|0.172/0.251|**0.171/0.250**|
> ||192|**0.237/0.293**|0.239/0.295|
> ||336|0.299/0.332|**0.296/0.331**|
> ||720|**0.391/0.387**|0.393/0.390|
> |Weather|96|0.156/**0.193**|**0.155**/0.194|
> ||192|**0.209/0.243**|**0.209/0.243**|
> ||336|**0.262/0.283**|0.264/**0.283**|
> ||720|**0.339/0.334**|0.342/**0.334**|
>
> These results show that LMA achieves comparable or better performance across the majority of settings, particularly on datasets with more nonstationary or high-frequency characteristics (e.g., ETTm2 and Weather). Even without access to fixed filtering kernels, the model is able to autonomously discover effective decomposition patterns, suggesting that LMA enhances adaptability while maintaining stability.
>
> ---
>
> ### **[Q4]: Can PMLF be extended beyond two components (trend + seasonal) to model noise, regime shifts, or multiple periodicities?**
> **Response.** We appreciate the reviewer’s insightful question. Indeed, the current formulation of PMLF focuses on a two-component decomposition (seasonal and trend), which aligns well with many standard forecasting benchmarks. However, the underlying philosophy of PMLF is structurally extensible. If the time series is decomposed into $n$ components (e.g., trend, multiple periodicities, noise, regime shifts), the framework can be readily generalized by assigning a dedicated loss to each component:
>
> $$\mathcal{{L}_{total}}=\sum_i^n \lambda_i \mathcal{L_i}$$
>
> This formulation remains fully compatible with our dynamic weighting strategy, enabling adaptive focus across components. Moreover, our use of physically guided loss design can be generalized, for instance, using entropy-like losses for noise components or change-point-aware losses for regime shifts.
>
> ### **[Q5]: Does the proposed log-based loss for trend modeling incur optimization issues (e.g., vanishing gradients) or slow convergence compared to MSE?**
>
> **Response:** We thank the reviewer for raising this important concern. To ensure training stability and computational efficiency, we carefully designed the logarithmic loss function to avoid numerical pitfalls and empirically evaluated its behavior during optimization.
>
> The trend loss $\mathcal{L}_\tau$ adopts a logarithmic form:
> $$\mathcal{L}\_\tau = \frac{1}{CH} \sum\_{c=1}^C\sum\_{h=1}^H \log(1 + | \hat{\tau}^{(c)}\_{T+h} - \tau^{(c)}\_{T+h} | )$$
>
> with gradient:
>
> $$
> \frac{d \mathcal{L}_{\tau}}{de} = \frac{1}{1 + e} \in (0,1]
> $$
>
> Although $\mathcal{L}_\tau$ is non-convex, it is Lipschitz-continuous and differentiable. Thus, the gradient vanishes gradually with growing error, which provides a soft constraint on large deviations while preserving stability during training. Therefore, the logarithmic trend term loss will not encounter optimization difficulties.
>
> Due to the inability to upload images, we provided the loss variation of each epoch validation set trained with different loss functions on ETTh2 using Amlfier
>
> |Epoch|1|2|3|4|5|6|7|8|9|10
> |-|-|-|-|-|-|-|-|-|-|-|
> |MSE|0.2183|0.2160|0.2150|0.2127|0.2146|0.2138|0.2139|0.2139|0.2140|0.2141
> |PMLF|0.1382|0.1364|0.1357|0.1352|0.1352|0.1351|0.1351|0.1351|0.1351|0.1351
>
> In summary, the log-based trend loss does not suffer from vanishing gradients or convergence slowdown, and instead provides smoother convergence, improved robustness, and more stable optimization.
>
> We sincerely thank the reviewers for their thorough and constructive feedback. We have carefully addressed all concerns regarding loss design, weight stability, learnable decomposition, and optimization behavior through both controlled experiments and theoretical analysis. Additional materials and clarifications will be provided in the final version to further enhance transparency and reproducibility.

---

> > ### Comment · Reviewer_eskT · 2025-08-06
> >
> > Thanks for the explanation and supplementary materials.
> > Basically, most questions are answered, while some points may go beyond this current work.
> > I would like to keep my current rating.

---

> ### Author Response · Authors · 2025-08-06
>
> Dear Reviewer eskT,
>
> We sincerely thank you for your time and thoughtful review.
>
> We have carefully addressed your comments in our rebuttal. If you happen to have any further thoughts or concerns, we would be more than happy to clarify and provide any additional information that might be helpful.
>
> We fully understand your busy schedule, and we truly appreciate your support in improving our work.
>
> Many thanks,
>
> 7051 Authors

---

### Official Review · Reviewer_6tYu · 2025-07-06

**Clarity:** 3
**Significance:** 2
**Originality:** 3
**Rating:** 4
**Confidence:** 3

**Summary:**

This work assumes that real-world time series can be decomposed into seasonal and trend components and proposes using different loss functions tailored to each. Seasonal components, which represent high-frequency oscillations, are modeled with a quadratic loss, capturing their need for local precision. Trend components, reflecting long-term, low-frequency changes, are modeled with a logarithmic loss to ensure global robustness. A softmax-based balancing strategy is introduced to adaptively weight these losses based on their energetic behavior. The proposed method outperform compared approach on multiple benchmarks

**Questions:**

- Can the authors provide empirical evidence (e.g., using toy or real-world datasets) to support the claim that unified loss functions fail to capture both trend and seasonal components effectively?
- Have the authors considered including ablation studies to evaluate whether the performance gains come from the decomposition itself or the choice of distinct loss functions?
- Have different combinations of loss functions (e.g., quadratic for both, logarithmic for both, or swapped) been tested to justify the specific assignment proposed?
- How does the proposed adaptive weighting compare against fixed or heuristic alternatives in terms of performance? Can a baseline be included for comparison?

**Ethical Concerns:**

["NO or VERY MINOR ethics concerns only"]

**Limitations:**

yes

**Quality:**

2

**Strengths And Weaknesses:**

1. **Unrealistic Decomposition Assumption**
   The method assumes that all time series can be neatly decomposed into seasonal and trend components. This is a strong and often invalid assumption for many real-world signals that include abrupt changes, irregularities, or structural breaks. The paper does not discuss how the method handles such scenarios, raising concerns about robustness and generalization.

2. **Misleading "Physics-Guided" Terminology**
   While the paper uses analogies from physics to motivate the choice of loss functions, it does not incorporate any actual physical modeling or constraints. Referring to the method as "physics-guided" is misleading and may confuse readers, especially in the context of Physics-Informed Neural Networks (PINNs), which are grounded in physical laws.

3. **Lack of Justification for Key Claims**
   - The claim that a unified loss function cannot effectively handle both seasonal and trend patterns is asserted without theoretical or empirical support. This should be backed by experiments (e.g., on toy datasets) comparing standard losses such as MSE, MAE, and DTW to illustrate their limitations.
   - The specific choice of quadratic loss for seasonality and logarithmic loss for trend appears ad hoc. There is no ablation study to verify whether these choices are optimal, or whether the performance gains come mainly from the decomposition itself.
   - The effectiveness of the proposed adaptive loss weighting strategy is not clearly demonstrated. A comparison with fixed or heuristic weighting schemes is necessary to justify its inclusion.

---

> ### Author Rebuttal · Authors · 2025-07-31
>
> ### **[Q1]:provide empirical evidence to support the claim that unified loss functions fail to capture both trend and seasonal components effectively.**
>
> **Response**. Thank you for your insightful and constructive suggestion. We fully agree that empirical evidence is crucial to support our claim.
>
> To support our claim, we conducted a controlled experiment on the well-known AirPassengers dataset, which exhibits both a clear seasonal pattern and a long-term upward trend. We trained a simple LSTM model under two settings: (1) using a unified MSE loss, and (2) using a decomposed loss. All other training configurations were held constant.
>
> Results showed that the decomposed MSE setup achieved better forecasting performance (final MSE = 0.02065vs. 0.01267). This confirms that a structure-aware loss provides a more effective supervisory signal than its unified counterpart, even when using the same loss form. We believe this demonstrates the practical importance of respecting structural heterogeneity in loss design.
>
> ---
>
> ### **[Q2, Q3]: Is the performance gain of PMLF primarily due to the decomposition process, or the specific assignment of quadratic and logarithmic loss functions? Have alternative loss combinations been tested to justify the proposed design?**
>
> **Response**. We appreciate the reviewer’s insightful questions. We agree that distinguishing the contribution of decomposition and loss design is crucial to validate our method’s effectiveness.
>
> To disentangle these factors, we conducted an ablation study on five benchmark datasets (ETTh1/2, ETTm1/2, Weather). We kept the backbone model fixed and only varied the loss strategies, comparing:
>
> - **Unified loss baselines**: Directly applying MSE, MAE, Huber, and DTW directly applied to raw series.
>
> - **Structure-aware variants**: applying decomposition with various seasonal-trend loss assignments, including:quadratic for both(Q+Q), logarithmic for both (L+L), or swapped (L + Q) and Ours(Q+L)
>
> Due to space constraints, we report average forecasting MSE/MAE across four prediction horizons (96/192/336/720) based on Amplifier. A subset of results is summarized below:
>
> | Loss|PMLF||MSE||Huber||DTW||MAE||Q+Q||L+L||L+Q||
> |---|---|---|---|---|---|---|---|---|---|---|---|---|---|---|---|---|
> |Metric|MSE|MAE|MSE|MAE|MSE|MAE|MSE|MAE|MSE|MAE|MSE|MAE|MSE|MAE|MSE|MAE|
> |ETTh1|**0.443**|**0.434**|0.464|0.447|0.475|0.445|0.463|0.443|0.459|0.447|0.480|0.453|0.455|0.436|0.481|0.451|
> |ETTh2|**0.356**|**0.385**|0.383|0.409|0.372|0.399|0.378|0.405|0.369|0.394|0.377|0.404|0.363|0.387|0.387|0.386|
> |ETTm1|0.391|**0.386**|**0.384**|0.398|0.388|0.392|0.389|0.399|0.394|**0.386**|0.390|0.399|0.396|0.387|0.389|0.398|
> |ETTm2|**0.275**|**0.316**|0.281|0.327|0.277|0.322|0.282|0.328|0.276|0.318|0.313|0.347|0.277|0.322|0.279|0.325|
> |Weather|**0.242**|**0.263**|0.247|0.274|0.244|0.268|0.245|0.273|0.243|**0.263**|0.248|0.274|0.245|0.268|0.248|0.274|
>
> Based on the above results, we draw the following conclusions:
> - The performance gain is not solely attributed to decomposition, as the decomposition-based methods （e.g Q+Q, L+L） alone do not consistently outperform unified loss functions across all datasets;
> - Our proposed loss combination (quadratic for seasonal and logarithmic for trend) consistently outperforms all other combination  variants, including Q+Q, L+L, and L+Q;
> - Therefore,  the observed performance gains arise from the synergy between a meaningful decomposition and a component-aware loss design.
>
> We hope this empirical analysis and the above insights sufficiently address your concern. Your thoughtful comments were highly valuable and helped us improve the clarity and rigor of our work.
>
> ---
>
> ### **[Q4]: How does the proposed adaptive weighting compare against fixed or heuristic alternatives?**
>
> **Response**. We appreciate the reviewer’s suggestion to evaluate the benefits of our adaptive weighting mechanism through direct comparison with fixed or heuristic strategies.
>
> To investigate this, we conducted ablation experiments on two representative datasets (ETTh2 and ETTm2), using the same model backbone (Amplifier). We compared our dynamic strategy (PMLF) with heuristic loss weight settings, where the seasonal-to-trend ratio ranged from 1:9 to 9:1.
>
> |loss strategy||dynamic strategy ||1:9||2:8||3:7||4:6||5:5||6:4||7:3||8:2||9:1||
> |---|---|---|---|---|---|---|---|---|---|---|---|---|---|---|---|---|---|---|---|---|---|
> Dataset|horizon|MSE|MAE|MSE|MAE|MSE|MAE|MSE|MAE|MSE|MAE|MSE|MAE|MSE|MAE|MSE|MAE|MSE|MAE|MSE|MAE|
> ETTh2|96|**0.283**|**0.332**|0.292|0.335|0.293|0.335|0.293|0.335|0.292|0.334|0.290|0.333|0.290|0.332|0.289|**0.332**|0.290|0.334|0.294|0.337
> ||192|**0.353**|0.379|0.358|0.379|0.357|0.379|0.358|0.379|0.356|0.378|0.354|**0.377**|0.356|0.378|0.359|0.380|0.363|0.381|0.370|0.384|
> ||336|**0.385**|0.405|0.390|0.407|0.386|**0.403**|0.392|0.408|0.388|0.405|0.400|0.409|0.387|0.403|0.392|0.406|0.394|0.408|0.400|0.411|
> ||720|**0.403**|**0.423**|0.407|0.426|0.420|0.434|0.420|0.433|0.405|0.500|0.424|0.425|0.405|0.425|0.406|0.425|0.409|0.427|0.413|0.429|
> ||Avg|**0.356**|**0.385**|0.362|0.387|0.364|0.388|0.366|0.389|0.360|0.404|0.367|0.386|0.360|0.385|0.362|0.386|0.364|0.388|0.369|0.390
> ||
> ETTm2|96|**0.172**|0.251|0.173|0.251|0.173|0.250|0.173|0.250|0.171|0.250|0.172|0.250|**0.171**|**0.249**|0.172|0.250|0.172|0.250|0.173|0.252|
> ||192|**0.237**|**0.293**|0.239|0.294|0.238|0.294|0.238|0.294|0.239|0.294|0.239|0.294|0.240|0.295|0.238|0.294|0.239|0.295|0.239|0.295|
> ||336|**0.299**|**0.332**|0.300|0.333|0.300|0.332|0.300|0.334|0.300|0.333|0.302|0.334|**0.299**|0.333|**0.299**|**0.332**|**0.299**|**0.332**|0.301|0.334|
> ||720|**0.391**|**0.387**|0.397|0.392|0.394|0.392|0.396|0.389|0.399|0.391|0.398|0.391|0.400|0.399|0.398|0.392|0.397|0.391|0.398|0.392|
> ||Avg|**0.275**|**0.316**|0.277|0.318|0.276|0.317|0.277|0.317|0.277|0.317|0.278|0.317|0.278|0.319|0.277|0.317|0.276|0.317|0.278|0.318|
>
> The results show that while certain fixed ratios (e.g., 7:3 on ETTm2) may yield comparable performance in specific horizons, no single fixed setting consistently outperforms the adaptive approach. In contrast, our method achieves the best or near-best results across all settings without manual tuning.
>
> While fixed heuristics can perform well in some cases, they lack adaptability and require manual tuning. In contrast, our softmax-based strategy dynamically learns to balance seasonal and trend losses in a data-driven manner, offering robust and generalizable performance across diverse scenarios. We appreciate the reviewer’s suggestion, which prompted us to conduct this comparative study and reinforced the value of our design.
>
> ---
>
> ### **[W1] The dilemma of decomposing assumptions**
> We thank the reviewer for highlighting this important point. Indeed, PMLF operates under the assumption that the time series can be approximately decomposed into a low-frequency trend and a high-frequency seasonal component. We explicitly acknowledge this limitation in Section 5 of the paper.
>
> However, the real-world datasets used in our experiments, like Weather and ETTm2, exhibit clear signs of imperfect separability, including irregular fluctuations, external shocks, and nonstationary behaviors. Despite this, PMLF still demonstrates strong performance across all benchmarks, which suggests practical robustness even under imperfect separability. To further mitigate the rigidity of decomposition,  we introduced a learnable moving average (LMA) mechanism, as shown in the last column of Table 3. Unlike fixed-kernel smoothing, LMA allows the decomposition boundary to be **learned** rather than fixed, enabling the model to dynamically adapt to structural irregularities and regime shifts in a data-driven manner.
>
> While we agree that perfect decomposition is not always feasible, the learnable decomposition strategy helps PMLF remain robust and adaptable in practice. We appreciate the reviewer for prompting us to emphasize this point more clearly.
>
> ---
>
> ### **[W2] Misleading "Physics-Guided" Terminology**
>
> We appreciate the reviewer’s concern regarding the terminology. In our work, the term “physics-guided” is used in a qualitative and analogical sense, rather than implying strict enforcement of physical laws or differential equations, as done in physics-informed neural networks (PINNs). Specifically, our seasonal loss adopts a quadratic form motivated by the potential energy of a harmonic oscillator (Eq. 3–7), while the trend loss uses a logarithmic form inspired by the diminishing structural response seen in irreversible relaxation systems (Eq. 8–10). These formulations are intended to provide interpretability and design intuition, rather than enforce hard physical constraints.
>
>  We understand how the term "physics-guided" may lead to unintended interpretations. To prevent confusion, we will revise the terminology throughout the paper and use “physics-inspired” instead, which more accurately reflects the nature and intent of our design.
>
> ---
>
> ### W3.3 Please refer to the answer Q4
> - **W3.1** Please refer to the response to Q1.
> - **W3.2** Please refer to the responses to Q2 and Q3.
> - **W3.3** Please refer to the response to Q4.

---

> > ### Author Response · Authors · 2025-08-06
> >
> > Dear Reviewer 6tYu,
> >
> > We sincerely thank you for your time and thoughtful review.
> >
> > We have carefully addressed your comments in our rebuttal. If you happen to have any further thoughts or concerns, we would be more than happy to clarify and provide any additional information that might be helpful.
> >
> > We fully understand your busy schedule, and we truly appreciate your support in improving our work.
> >
> > Many thanks,
> >
> > 7051 Authors

---

> > > ### Comment · Reviewer_6tYu · 2025-08-07
> > > **Response to Reviewers**
> > >
> > > Thank you to the reviewers for their thorough feedback. Although the authors have addressed my initial concerns, their central decompaction claim—that each decomposed component requires its specific loss—remains unsupported by the current results. In the ablation table, the baseline ordering (L + Q) matches—or even slightly underperforms—the proposed reversed sequence (Q + L), directly contradicting the authors’ assertion. This marginal, opposite trend undermines the rationale for assigning distinct losses to each component.
> > >
> > > Therefore, I will still keep my scores.

---

> > > > ### Author Response · Authors · 2025-08-08
> > > >
> > > > Dear Reviewer  6tYu,
> > > >
> > > > As the discussion phase is approaching its end, we would like to kindly ensure that our previous clarification regarding your concern has been received. We sincerely appreciate your feedback and hope our response was helpful.
> > > >
> > > > Thank you again for your time and consideration.
> > > >
> > > > Best regards,
> > > >
> > > > 7051 Authors

---

> > > ### Author Response · Authors · 2025-08-07
> > >
> > > **Clarification.** Thank you for your thoughtful response. We realize that the labeling in the ablation table may have caused confusion. To clarify, our proposed PMLF corresponds to the Q+L configuration, where the seasonal component uses quadratic loss and the trend component uses log-style loss. The L+Q setting was added later based on your suggestion. As shown in the table below, PMLF (Q+L) consistently outperforms Q+Q, L+L, and L+Q, which supports our design motivation.
> > >
> > > | Loss|PMLF(Q+L)||Q+Q||L+L||L+Q||
> > > |---|---|---|---|---|---|---|---|---
> > > |Metric|MSE|MAE|MSE|MAE|MSE|MAE|MSE|MAE|MSE|
> > > |ETTh1|**0.443**|**0.434**|0.480|0.453|0.455|0.436|0.481|0.451|
> > > |ETTh2|**0.356**|**0.385**|0.377|0.404|0.363|0.387|0.387|0.386|
> > > |ETTm1|0.391|**0.386**|0.390|0.399|0.396|0.387|**0.389**|0.398|
> > > |ETTm2|**0.275**|**0.316**|0.313|0.347|0.277|0.322|0.279|0.325|
> > > |Weather|**0.242**|**0.263**|0.248|0.274|0.245|0.268|0.248|0.274|
> > >
> > > Note: Lower MSE and MAE values indicate better forecasting performance.
> > >
> > > We hope this clarification helps and truly appreciate your careful examination of our work.

---

### Note · Authors · 2025-08-12

We sincerely thank all reviewers and the AC for their constructive feedback, which has greatly helped us improve the clarity, completeness, and rigor of our work. We are pleased that the reviewers recognized the following strengths:
- **A physics-inspired, component-specific loss framework with clear theoretical motivation.** (All Reviewers)
- **Model-agnostic design** that integrates seamlessly into diverse forecasting architectures. (Reviewer eskT, jE1W)
- **Consistent performance gains** across datasets, models, and baselines. (All Reviewers)
- **An effective dynamic weighting strategy** enabling stable and balanced optimization. (Reviewer eskT, mT8C, jE1W)

Main concerns and our responses:
- **Effectiveness of the proposed seasonal/trend loss design.** In response to concerns from Reviewers 6tYu, eskT, and jE1W, we performed a full ablation comparing Q+L (PMLF) with Q+Q, L+L, and L+Q. **PMLF consistently achieved the best results**, addressing this point for most reviewers, though Reviewer 6tYu misinterpreted Q+L as L+Q.

- **Evaluation of alternative loss functions.** Following suggestions from Reviewers eskT and jE1W, we evaluated MAE, Huber, DWT, and others, all of which underperformed PMLF. This resolved the concern without further questions.

- **Generality across datasets.** To ensure robustness, we extended experiments to seven benchmarks, where PMLF consistently outperformed other structure-aware losses. No follow-up questions remained.

- **Clarification of definitions and theoretical design.** Reviewer eskT requested clearer definitions and deeper discussion of the theoretical design. We added **formal reasoning, supporting ablations, and expanded theoretical context**, strengthening the rigor and clarity of our presentation. No further concerns were expressed.

- **Computational complexity and convergence.** Reviewers eskT and mT8C inquired about computational overhead and convergence behavior. We provided **theoretical proof of convergence** and experimental evidence showing stable convergence without significant complexity costs, which met the reviewers’ expectations.

Through these targeted experiments, additional analyses, and manuscript revisions, **we believe we have fully addressed all substantive reviewer concerns. Our physics-guided, component-specific loss framework is theoretically grounded, empirically validated, model-agnostic, and interpretable, representing a valuable contribution to time series forecasting.**

---

### Decision · Program_Chairs · 2025-09-17

**Decision:**

Accept (poster)

**Comment:**

The paper introduces a Physics-guided Multiscale Loss Framework (PMLF) for time series forecasting. The approach decomposes time series into seasonal and trend components and applies distinct loss functions: a quadratic loss for seasonal fluctuations and a logarithmic loss for long-term trends, drawing inspiration from analogies in molecular vibration and drift energies. An adaptive softmax-based weighting balances these two losses dynamically. The framework is designed to be model-agnostic, and experiments across multiple benchmarks show consistent improvements in forecasting accuracy.

Strengths:  This paper addresses an interesting issue in time series forecasting --- unified losses often fail to capture heterogeneous time series dynamics. The proposed framework is lightweight, model-agnostic, and easy to integrate into existing forecasting methods. Empirical evaluations span several datasets and backbone architectures, with improvements that are often consistent and practically relevant.

Weaknesses:  A major concern about the proposed loss function modeling is that the physics analogy is metaphorical rather than strictly grounded, which reduces the conceptual novelty. In addition, some experiments (e.g., oscillations in trend predictions) are not fully explained, and additional comparisons with structure-aware baselines would strengthen the evaluation. The adaptive weighting mechanism remains under-analyzed, lacking discussion of stability or convergence properties.

Rebuttal: The authors have provided extensive experiments and detailed clarification to the concerns. Ablation studies on the choices of loss functions and other heuristic alternatives for the weighting of loss functions. A better interpretation of the physical meaning the designs is also provided.

In summary, this paper is a borderline paper. Reviewers raised concerns about the limited novelty, the metaphorical nature of the physics framing, and unclear benefits of the adaptive weighting. In the rebuttal, the authors provided further ablation studies, clarifying implementation choices and showing that the adaptive scheme contributes positively. While some skepticism remains, especially regarding the theoretical grounding, the additional evidence helped mitigate the initial concerns. On balance, I find the empirical strengths and practical utility outweigh the limitations, leading to a positive but cautious recommendation.